# Graph Tokenization for Bridging Graphs and Transformers

**Zeyuan Guo** [*]
Beijing University of Posts and Telecom.,
Beijing 100876, China
guozeyuan@bupt.edu.cn

**Enmao Diao** [*]
DreamSoul
enmao.diao@dreamsoul.com

**Cheng Yang**
Beijing University of Posts and Telecom.,
Beijing 100876, China
yangcheng@bupt.edu.cn

**Chuan Shi** [†]
Beijing University of Posts and Telecom.,
Beijing 100876, China
shichuan@bupt.edu.cn

## Abstract

The success of large pretrained Transformers is closely tied to tokenizers, which convert raw input into discrete symbols. Extending these models to graph-structured data remains a significant challenge. In this work, we introduce a graph tokenization framework that generates sequential representations of graphs by combining reversible graph serialization, which preserves graph information, with Byte Pair Encoding (BPE), a widely adopted tokenizer in large language models (LLMs). To better capture structural information, the graph serialization process is guided by global statistics of graph substructures, ensuring that frequently occurring substructures appear more often in the sequence and can be merged by BPE into meaningful tokens. Empirical results demonstrate that the proposed tokenizer enables Transformers such as BERT to be directly applied to graph benchmarks without architectural modifications. The proposed approach achieves state-of-the-art results on 14 benchmark datasets and frequently outperforms both graph neural networks and specialized graph transformers. This work bridges the gap between graph-structured data and the ecosystem of sequence models. Our code is available at here.

## 1 Introduction

Large pretrained Transformer models (Vaswani et al., 2017; Minaee et al., 2024), exemplified by LLMs, have achieved state-of-the-art results across diverse domains (Dosovitskiy et al., 2020; Gong et al., 2021). A key component of this success is the tokenizer, which converts raw input into sequences of discrete symbols. By structuring information into learnable units, the tokenizer provides the interface between complex data and Transformer architectures, supporting the scalability and performance of these models.

Research on extending Transformers to graph-structured data has explored two main strategies, each with inherent limitations (Yu et al., 2025; Guo et al., 2024). One strategy modifies the architecture by incorporating attention mechanisms into Graph Neural Networks (GNNs) to create specialized Graph Transformers (Yun et al., 2019). These approaches require graph-specific designs that diverge from standard sequence models and their ecosystem. The other strategy converts graphs into continuous embeddings for use with Transformers (Tang et al., 2024), but this often causes information loss or unstable representations, which can degrade model performance (Chen et al., 2024).

Developing a principled graph tokenizer requires reexamining the notion of tokenization in the context of graph-structured data. Specifically, text can be modeled as a path graph, where the linear sequence of tokens provides both a fixed neighborhood structure and a canonical ordering, making tokenization relatively straightforward. In contrast, general graphs pose additional challenges, as

---

[*]Equal Contribution
[†]Corresponding author

their neighborhoods can branch in multiple directions rather than follow a simple linear sequence. They also lack permutation invariance, where graphs under node permutations are considered equivalent. Furthermore, co-occurrence statistics widely used in text, such as n-gram frequencies based on contiguous tokens, are not directly applicable to graphs.

We propose a framework that addresses these challenges by integrating graph serialization with Byte Pair Encoding (BPE), a data-driven compression algorithm widely applied in text tokenization (Shibata et al., 1999). To ensure that graph structure and labels are preserved, we adopt reversible serialization methods such as extended Euler circuits and minimal-weight graph traversals. Ordering ambiguity is resolved by using global statistics to deterministically guide the serialization process, which translates common substructures into frequent and adjacent symbol patterns that BPE is well suited to merge. Specifically, BPE iteratively merges the most frequent pairs of symbols into new tokens, thereby reducing sequence length while preserving common substructures. As a result, applying BPE to serialized graphs enables the construction of a vocabulary of frequent graph neighborhoods, producing discrete tokens that are both informative and well aligned with Transformer architectures.

In this work, our contributions can be summarized as follows:

- **General Framework for Graph Tokenization.** We introduce a tokenization framework that combines reversible graph serialization with BPE. By decoupling the encoding of graph structure from the model architecture, this framework provides an effective interface that enables standard off-the-shelf Transformer models to be applied directly to graph-structured data without requiring any architectural modifications.

- **Structure-Guided Serialization for BPE.** We propose a deterministic serialization process guided by global statistics of graph substructures. The process addresses ordering ambiguities in graphs and aligns frequently occurring substructures into adjacent sequence patterns. Structure-Guided Serialization provides an effective basis for BPE to learn a meaningful and interpretable vocabulary of structural graph tokens.

- **State-of-the-Art Performance on Downstream Tasks.** Our tokenizer enables standard Transformer backbones to achieve state-of-the-art results across a diverse suite of 14 benchmarks for graph classification and regression. The proposed approach frequently outperforms both established Graph Neural Networks and specialized Graph Transformers, demonstrating its effectiveness and generalization.

## 2 RELATED WORKS

**Graph Neural Networks.** Graph Neural Networks (Kipf, 2016; Luo et al., 2025) are the prevailing framework for learning on graph-structured data. They rely on message passing, where node representations are updated by iteratively aggregating information from local neighbors, enabling effective modeling of local graph structure (Chen et al., 2019). To capture dependencies beyond local neighborhoods, subsequent work introduced self-attention, leading to Graph Transformers (Yun et al., 2019; Wu et al., 2023) and hybrid global-local models (He et al., 2023; Zhang et al., 2023). More recently, graph representation learning has been combined with Graph Foundation Models, often by mapping graph structure and features into the embedding space of pretrained foundation models (Tang et al., 2024; Chen et al., 2024; Zhang et al., 2025; Yan et al., 2025; Liu et al., 2026). These approaches depend on cross-modal alignment, with performance influenced by the semantic compatibility between graph features and natural language. Our objective is to design an interface that enables graphs to be processed directly by standard, off-the-shelf Transformers.

**Graph Serialization.** Serialization of a graph into a sequence was one of the earliest strategies for applying sequence-based models. Early methods such as DeepWalk generated node sequences through random walks and processed them with shallow neural networks (Perozzi et al., 2014; Zhang et al., 2020). This direction was later surpassed by the message passing paradigm of GNNs (Gilmer et al., 2017), which became the dominant approach to learning graph representations. More recently, the success of sequence-native architectures such as the Transformer has renewed interest in serialization-based methods (Wang et al., 2024). Many existing graph-to-sequence pipelines are not reversible. Specifically, walk-based serializations break the graph into local fragments. Each sequence reflects only part of the graph, and even combining many walks cannot reconstruct the original structure or capture global connectivity (Xia et al., 2019). In another case, traversal-based serializations are sensitive to node ordering and starting choices, so even isomorphic graphs may

produce different graph traversal circuits (Gao et al., 2025). In contrast, our method is reversible and almost invariant to graph permutation.

**Tokenization** The Transformer architecture has become the standard paradigm for sequence modeling (Vaswani et al., 2017). Its success is closely tied to the use of effective tokenization (Floridi & Chiriatti, 2020; Guo et al., 2025), which is especially critical in LLMs. A tokenizer converts raw input (e.g., text) into a sequence of discrete symbols, with BPE being a widely adopted data-driven approach that builds a vocabulary by iteratively merging frequent symbol pairs (Shibata et al., 1999). In prior work on graph data, the term *graph tokenization* has been used with different meanings. It has referred to neural encoders that produce continuous embeddings (Tang et al., 2024), pooling or coarsening modules that compress subgraphs into super-nodes (Shen & Póczos, 2024), and vector quantization components that discretize node features or latent representations (Yang et al., 2023). In this paper, we adopt the standard definition in natural language processing, where a tokenizer is a deterministic procedure that maps a labeled graph to a sequence of discrete symbols for direct use by sequence models, in contrast to neural approaches (e.g., PS-VAE (Kong et al., 2022)) that project substructures into continuous latent spaces via learnable encoders (see Appendix B.1 for a detailed comparison).

## 3 METHOD

### 3.1 PRELIMINARIES

**Graph.** A graph is a tuple $G = (\mathcal{V}, \mathcal{E})$, composed of a finite set of nodes $\mathcal{V}$ and a set of edges $\mathcal{E}$. Our work focuses on *labeled graphs*, which we define as a tuple $\mathcal{G} = (G, L, \Sigma)$, where $\Sigma$ is a finite alphabet of symbols and $L : \mathcal{V} \cup \mathcal{E} \to \Sigma$ is a labeling function. Two labeled graphs $\mathcal{G}_1 = (G_1, L_1, \Sigma)$ and $\mathcal{G}_2 = (G_2, L_2, \Sigma)$ are *isomorphic*, denoted $\mathcal{G}_1 \cong \mathcal{G}_2$, if there exists a graph isomorphism $\phi : \mathcal{V}_1 \to \mathcal{V}_2$ between $G_1$ and $G_2$ that also preserves all labels.

**Graph Serialization.** In general, a graph serialization function $f$ maps a graph to a finite sequence of symbols. Let $\mathcal{A}$ denote the universe of possible sequence elements. The mapping is defined as

$$f : \mathcal{G} \mapsto (s_1, s_2, \ldots, s_k) \quad \text{s.t. } s_i \in \mathcal{A} \text{ for } 1 \leq i \leq k. \tag{1}$$

The choice of $\mathcal{A}$ depends on the serialization method. It may consist of node identifiers ($\mathcal{A} = \mathcal{V}$), continuous embeddings ($\mathcal{A} = \mathbb{R}^d$), or discrete labels ($\mathcal{A} = \mathbb{Z}$). For the purpose of building a discrete tokenizer, we focus on serializations where the output sequence is composed of symbols from the graph's alphabet, i.e., $\mathcal{A} = \Sigma$. To serve as a reliable interface, such a serialization should satisfy two key properties:

- **Reversibility.** A serialization $f$ is reversible if the original labeled graph $\mathcal{G}$ can be recovered from its sequence $S = f(\mathcal{G})$ up to isomorphism. Formally, let $f^{-1}(S)$ denote the set of all graphs that could produce sequence $S$. The serialization is reversible if for any $\mathcal{G}$ in the domain of $f$, there exists a reversed graph $\mathcal{G}' \in f^{-1}(f(\mathcal{G}))$ such that $\mathcal{G}' \cong \mathcal{G}$.

- **Determinism.** A serialization function $f$ is deterministic if, for any labeled graph $\mathcal{G}$, it consistently produces the same sequence $S$. This property is essential for addressing the permutation-invariance of graphs. A deterministic serialization generates a stable sequence for all graphs within an isomorphism class.

**Graph Tokenization.** A graph tokenizer $\Phi$ maps a labeled graph $\mathcal{G}$ to a finite sequence of discrete symbols, referred to as tokens.

$$\Phi : \mathcal{G} \mapsto S_{\mathrm{T}} = (t_1, \ldots, t_m), \quad t_j \in \mathcal{V}_T. \tag{2}$$

In this work, we construct the graph tokenizer $\Phi$ by composing a graph serialization function $f$ with a sequence tokenizer $T$ inspired by the text tokenizers used in LLMs. The sequence tokenizer $T$ maps a sequence over the initial alphabet $\Sigma$ to a new sequence over a target vocabulary $\mathcal{V}_T$, where the vocabulary is typically learned from data using BPE. The overall mapping is given by

$$\Phi = T \circ f. \tag{3}$$

When a decoding procedure is available, the original graph can be reconstructed up to isomorphism (i.e., preserving complete topology and all labels; strict node-index identity requires an additional index mapping) by applying the inverse operations $T^{-1}$ followed by $f^{-1}$. Specifically, the term *graph tokenizer* refers to methods that produce a discrete sequence. Methods that only discretize embeddings (Yang et al., 2023) or apply pooling or coarsening (Shen & Póczos, 2024) are not considered tokenizers in this sense.

## 3.2 GRAPH TOKENIZER

We construct our graph tokenizer $\Phi$ by composing a reversible and structure-guided serialization function $f$ with a tokenization step $T$ based on BPE. To ensure graph structural information is preserved, $f$ is designed to be *reversible*, and to produce stable sequences, we enforce a *deterministic* guiding policy for $f$. We propose a data-driven graph tokenizer $\Phi$ that is learned from a training corpus of graphs rather than relying on hand-crafted heuristics. Specifically, Algorithm 1 details the training, encoding, and decoding procedures of `GraphTokenizer`, and Figure 1 illustrates the overall framework.

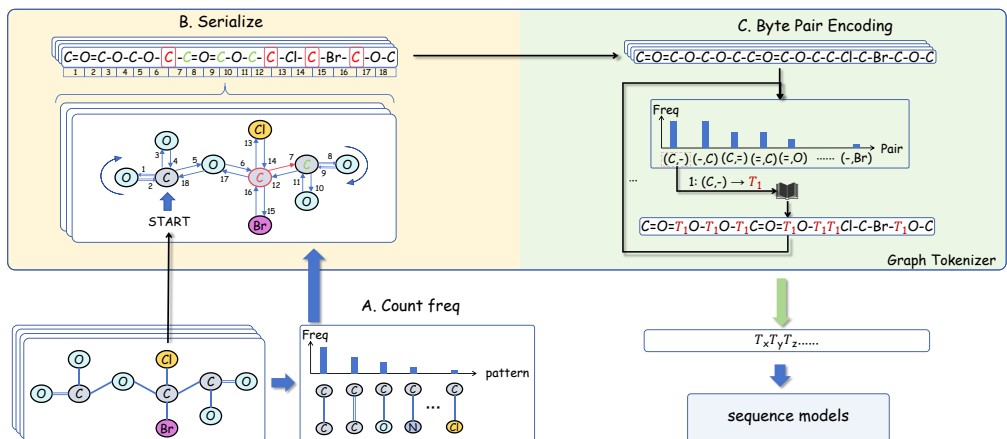

Figure 1: Framework of the proposed graph tokenizer. (A) Substructure frequencies are collected from the training graphs. (B) Structure-guided and reversible serialization is performed using a frequency-guided Eulerian circuit, where the next edge is selected according to a priority rule (e.g., red C: 7→13→15→17). (C) A BPE vocabulary is trained on the serialized corpus, and graphs are encoded into discrete tokens for use in downstream sequence models.

**Local Structural Pattern Statistics.** The training procedure begins with the computation of dataset-level statistics of local patterns, which provide a data-driven basis for ensuring determinism in graph serialization. As illustrated in Fig. 1A, we count how often small labeled patterns appear in the training graphs, using molecular graphs and edge patterns as an example. These counts are then normalized into relative frequencies.

For a labeled graph $\mathcal{G} = ((\mathcal{V}, \mathcal{E}), L, \Sigma)$, we define a basic local pattern as a labeled edge $p = (l_u, l_e, l_v) \in \Sigma^3$, which captures the labels of the source node, the edge, and the target node. Intuitively, this is the smallest substructure that still reflects a typed relation between two labeled entities. Compared with larger subgraphs, it is computationally inexpensive, permutation-invariant over node indices, and stable under isomorphisms, which makes it a practical choice for tie-breaking during graph serialization. The count of $p$ in $\mathcal{G}$ is given by

$$\text{Count}(\mathcal{G}, p) = \big| \{ e = (u, v) \in \mathcal{E} \ | \ (L(u), L(e), L(v)) = p \} \big|. \tag{4}$$

Aggregating over the training set $\mathcal{D}$, we obtain raw counts and their normalized relative frequencies as

$$C(p) = \sum_{\mathcal{G} \in \mathcal{D}} \text{Count}(\mathcal{G}, p) \qquad F(p) = \frac{C(p)}{\sum_{p' \in \Sigma^3} C(p')}. \tag{5}$$

$F(p)$ denotes the normalized relative frequency, while raw counts $C(p)$ are introduced here as an intermediate definition.

**Structure-Guided Reversible Serialization.** We proceed to the next step of our framework, where each graph is converted into a sequence of symbols (Algorithm 1, **line 5**). In this procedure, line 4 corresponds to estimating $F$ from data, and the resulting frequency map guides the structure-aware serialization function $f_g(\cdot, F)$. The function $f_g$ addresses traversal ambiguities by prioritizing edges whose incident labeled pattern has higher $F(p)$ from itself and neighbors, with fixed lexical rules applied to break any remaining ties.

Fig. 1B illustrates this serialization process on a molecular graph. Specifically, at the red node C, the next step is chosen by the $F$-guided priority rather than arbitrarily (e.g., choose the green node C be-

Table 1: Properties of graph serialization methods. For Random Walk, $L$ is walk length and $R$ the number of walks. Implementation details are in Appendix B.

| Method | Reversibility | Determinism | Time Complexity |
|---|---|---|---|
| Random Walk | No | No | $O(RL)$ |
| Node-list BFS/DFS | No | No | $O(|\mathcal{V}| + |\mathcal{E}|)$ |
| Topological Sort | No | No | $O(|\mathcal{V}| + |\mathcal{E}|)$ |
| Eulerian circuit | Yes | No | $O(|\mathcal{E}|)$ |
| SMILES (non-canonical) | Yes | No | $O(|\mathcal{V}| + |\mathcal{E}|)$ |
| Canonical SMILES | Yes | Yes | $O(|\mathcal{V}| + |\mathcal{E}|)$ |
| Chinese Postman Problem | Yes | No | $O(|\mathcal{V}|^3)$ |
| **Frequency-Guided Eulerian circuit** | **Yes** | **Yes** | $O(|\mathcal{E}|)$ |
| **Frequency-Guided CPP** | **Yes** | **Yes** | $O(|\mathcal{V}|^3)$ |

cause "C-C" pair has the highest frequency). To ensure that graph serialization yields faithful graph representations, we require it to satisfy the properties of reversibility and determinism discussed in Section 3.1. Ensuring that these two properties hold simultaneously for general graphs is a significant challenge. To motivate our approach, we first review classical serialization methods against these criteria. To provide a clear overview, Table 1 summarizes the properties of existing serialization methods. A key limitation is that no classical method is simultaneously broadly applicable, fully reversible, and inherently deterministic.

Early approaches such as *Random Walks* (Perozzi et al., 2014) are inherently stochastic and typically explore only a local portion of the graph per sample. Even when many walks are aggregated, substructures are split across sequences without markers, so reconstruction is not guaranteed, and the procedure remains non-deterministic. Standard traversal algorithms like *Breadth-First Search (BFS)* (Moore, 1959) and *Depth-First Search (DFS)* (Even & Even, 2011) also fail to meet these requirements. Their non-determinism arises from arbitrary neighbor selection when multiple choices are available, and their node-list output omits edge connectivity, which prevents reconstruction of the original graph. *Topological Sort* (Kahn, 1962), which produces linear orderings for Directed Acyclic Graphs (DAGs), is limited to DAGs and admits multiple valid orderings, making it non-deterministic. Moreover, like other node-list traversals, it is not reversible because the precise edge connectivity information necessary for reconstruction is discarded.

In contrast to node-based traversals, methods that cover every edge of the graph are naturally reversible. A representative example is the *Eulerian circuit* (Biggs et al., 1986), which visits each edge exactly once. By treating each undirected edge as two opposing directed edges, the method can extend to any connected graph (Gao et al., 2025). During traversal, emitting an alternating *node–edge–node* sequence ensures that adjacent symbols share an endpoint, which preserves the information needed by $f^{-1}$ to reconstruct the edges. Despite this reversibility, the method remains non-deterministic because the classical Hierholzer's algorithm (Hierholzer & Wiener, 1873) must make arbitrary choices whenever a node has multiple unvisited edges. A related approach is the *Chinese Postman Problem (CPP)* (Kwan, 1960), which seeks a minimum weight traversal that covers all edges and thereby also preserves the complete graph structure. Non-determinism in CPP, although more constrained, is intrinsic to its solution process, typically solved using Edmonds' blossom algorithm (Edmonds & Johnson, 1973). The standard procedure first identifies all odd-degree nodes and constructs an auxiliary complete graph on them, with edge weights representing shortest path distances in the original graph. A minimum-weight perfect matching is then computed to determine which paths should be duplicated. If multiple minimum-weight matchings exist, the selection between them is arbitrary, which dictates the different possible traversals.

Domain-specific serialization methods for molecular graphs, such as SMILES (Weininger, 1988), represent a widely adopted approach in cheminformatics. Non-canonical SMILES is reversible but not deterministic, whereas canonical SMILES achieves determinism by applying an explicit canonicalization procedure under a fixed scheme. These procedures rely on chemistry-specific perception rules (e.g., aromaticity, implicit hydrogens, and structural notations) and therefore do not generalize to arbitrary labeled graphs. Furthermore, the determinism of canonical SMILES is defined relative to the chosen canonicalization algorithm and perception rules, and implementations may differ slightly across toolkits.

---

**Algorithm 1** The `GraphTokenizer` Framework

1: **Procedure** TRAIN($\mathcal{D}, K$)
2: **Input:** A training graph dataset $\mathcal{D}$; number of BPE merges $K$.
3: **Output:** frequency map $F$; BPE codebook $\mathcal{C} = (\mathcal{V}_T, \mathcal{R})$.
4: $F(p) \leftarrow \sum_{\mathcal{G} \in \mathcal{D}} \text{Count}(\mathcal{G}, p), \quad \forall p \in \Sigma^3$
5: $\mathcal{D}_S \leftarrow \{ f_g(\mathcal{G}, F) \mid \mathcal{G} \in \mathcal{D} \}$
6: $\mathcal{V}_T \leftarrow \Sigma; \ \mathcal{R} \leftarrow \emptyset$
7: **for** $k = 1$ to $K$ **do**
8: $\quad (s_a^*, s_b^*) \leftarrow \arg\max_{(s_a, s_b)} \sum_{S \in \mathcal{D}_S} \text{Count}(S, (s_a, s_b))$
9: $\quad s_{\text{new}} \leftarrow s_a^* \cdot s_b^*$
10: $\quad \mathcal{V}_T \leftarrow \mathcal{V}_T \cup \{s_{\text{new}}\}$
11: $\quad \mathcal{R} \leftarrow \mathcal{R} \cup \{(s_a^*, s_b^*) \rightarrow s_{\text{new}}\}$
12: $\quad$ **for** each $S \in \mathcal{D}_S$ **do**
13: $\quad\quad$ replace all disjoint adjacent pairs $(s_a^*, s_b^*)$ in $S$ with $s_{\text{new}}$
14: **return** $(F, \mathcal{C})$
15:
16: **Procedure** ENCODE($\mathcal{G}, F, \mathcal{C}$)
17: **Input:** graph $\mathcal{G}$; frequency map $F$; the codebook $\mathcal{C} = (\mathcal{V}_T, \mathcal{R})$.
18: **Output:** A token sequence $S_T$.
19: $S \leftarrow f_g(\mathcal{G}, F)$
20: **for** each $(s_a, s_b) \rightarrow s_{\text{new}}$ in $\mathcal{R}$ **do**
21: $\quad$ replace all disjoint adjacent pairs $(s_a, s_b)$ in $S$ with $s_{\text{new}}$
22: $S_T \leftarrow S$
23: **return** $S_T$
24:
25: **Procedure** DECODE($S_T, \mathcal{C}, f^{-1}$)
26: **Input:** token sequence $S_T$; codebook $\mathcal{C} = (\mathcal{V}_T, \mathcal{R})$; inverse serialization function $f^{-1}$.
27: **Output:** A reconstructed graph $\widehat{\mathcal{G}}$.
28: $S \leftarrow S_T$
29: **for** each $(s_a, s_b) \rightarrow s_{\text{new}}$ in reversed($\mathcal{R}$) **do**
30: $\quad$ replace every $s_{\text{new}}$ in $S$ with the pair $(s_a, s_b)$
31: $\widehat{\mathcal{G}} \leftarrow f^{-1}(S)$
32: **return** $\widehat{\mathcal{G}}$

---

Building on the preceding analysis, our strategy is to impose determinism on traversal methods that are inherently reversible. This is accomplished by introducing a guiding mechanism that leverages the global frequency map $F$ to resolve traversal ambiguities. In this way, we obtain a structure-guided graph serialization function $f_g$ that simultaneously satisfies reversibility and determinism for general graphs.

*Frequency-Guided Eulerian circuit* adapts Hierholzer's algorithm by introducing a priority rule. At any node $u$ with unvisited outgoing edges $\mathcal{E}_u$, the next edge $e^*$ is selected deterministically as

$$e^* = \arg\max_{e_i \in \mathcal{E}_u} \pi(e_i, F), \tag{6}$$

where $\pi(e_i, F)$ assigns a scalar priority, for example $\pi(e_i, F) = F(p_i)$ for the pattern $p_i = (L(u), L(e_i), L(v))$. Although traversal may begin from any node, the resulting circuit differs only by a cyclic shift.

For example, in Fig. 1B, when the traversal reaches the red C, there are four candidate neighbors (including the incoming one). According to the dataset-level statistics $F$, the C–C labeled-edge pattern has the highest $F(p)$, so $f_g$ takes that step (step 3). When it later returns to the same red C, it selects among the remaining three neighbors: the edge to Cl has the next-highest $F(p)$ (step 5), followed by steps 7 and 9.

*Frequency-Guided CPP* incorporates frequency statistics into the edge weights used by the solver. For an edge $e$ with associated pattern $p_e$, the weight is defined as

$$w(e) = \alpha \cdot 1 + (1 - \alpha) \cdot g(F(p_e)), \tag{7}$$

where $g$ is a decreasing function of frequency (e.g., $1/F(p_e)$) and $\alpha \in [0, 1]$ is a tunable hyperparameter. Ties that arise during matching or tour construction are resolved using the priority policy specified in Eq. 6. For disconnected graphs, each component is serialized independently and the results are concatenated in a fixed order.

**Vocabulary Learning via BPE.** After converting the graph dataset $\mathcal{D}$ into a corpus of symbol sequences $\mathcal{D}_S$, the final stage of training is to learn a vocabulary from this corpus. We employ Byte Pair Encoding (BPE), inspired by the text tokenizers used in LLMs, corresponding to the main loop in Algorithm 1 (**lines 6–14**). BPE iteratively identifies the most frequently occurring adjacent pair of symbols in the corpus and merges it into a new symbol added to the vocabulary. Fig. 1C illustrates the vocabulary learning procedure on a serialized molecular sequence. In this example, a pair denotes an adjacent atom–bond symbol, e.g. $(\mathrm{C}, -)$. At each iteration $i$, the most frequent pair is replaced at all disjoint occurrences by a new token $T_i$, and the corresponding merge rule $(s_a, s_b) \to T_i$ is added to the codebook $\mathcal{C}$. The updated sequence is then passed back to the counting step, forming an iterative training loop.

The key insight of our framework lies in the interplay between structure-guided serialization and the BPE algorithm. The serialization function $f_g$ is not merely a format conversion tool but leverages the global frequency map $F$ to ensure that statistically common local graph structures are systematically encoded as frequently adjacent symbol pairs in the sequence corpus $\mathcal{D}_S$. This structured corpus forms an ideal input for BPE's greedy merging strategy. When BPE merges the most frequent pair $(s_a^*, s_b^*)$ (**line 8**), the operation is not arbitrary compression but the discovery of statistically salient tokens derived from graph data. Each merged token represents a larger subgraph fragment that can be recovered from the serialization. The resulting vocabulary $\mathcal{V}_T$ provides a data-driven, structurally informed representation of the graph for the downstream Transformer.

**Encoding and Decoding.** After training, the procedure produces two components for inference: the frequency map $F$ and the BPE codebook $\mathcal{C}$. To encode a new graph, the ENCODE procedure in Algorithm 1 is applied. The graph is first serialized into a symbol sequence by the function $f$, we apply the merge rules $\mathcal{R}$ from $\mathcal{C}$ in the learned order to obtain the final token sequence $S_T$. The DECODE procedure in Algorithm 1 reverses this process. The tokens in $S_T$ are first expanded back into the original symbol sequence by applying the inverse of $\mathcal{R}$, and the inverse serialization function $f^{-1}$ then reconstructs the graph. These procedures ensure that the mapping between graphs and sequences is both reversible and deterministic, providing a bidirectional interface between the two domains.

**Applications.** The primary output of our framework is a discrete sequence of tokens $S_T$ that faithfully encodes the original graph. This sequential representation provides an interface through which the Transformer ecosystem can be directly applied to graph-structured data (Vaswani et al., 2017). For graph-level prediction tasks such as classification or regression, the token sequence can be processed by an encoder-only model (e.g., BERT). A special `[CLS]` token may be prepended, or the final hidden states pooled, to derive a vector representation for the entire graph (Perozzi et al., 2014). For generative tasks, a decoder-only model (e.g., GPT) can be trained to generate graphs autoregressively by predicting the next token in the sequence, supporting applications such as molecular or material discovery (Radford et al., 2019). Multimodal models can also support tasks such as graph summarization, where we use pretrained graph representations from the proposed tokenizer with a large language model to generate concise descriptions of input graphs (Yamagata et al., 2023).

In summary, the proposed graph tokenizer reframes graph representation learning as a sequence modeling problem. Our method decouples the structural complexity of graphs from the architectural design of the model and enables direct use of advances in sequence modeling, such as longer context windows (Ding et al., 2022) and more efficient attention mechanisms (Dao et al., 2022) for a wide range of graph learning tasks.

## 4 EXPERIMENTS

In this section, we evaluate our proposed graph tokenizer, `GraphTokenizer` (GT). We aim to answer the following questions: (1) How effectively does BPE compress the serialized graph representations, and what is the efficiency of our approach in terms of sequence length, processing speed, and training throughput? (2) How does our framework, when paired with standard Transformer models, perform against state-of-the-art graph representation learning methods? (3) How do different design choices, such as the serialization method and BPE usage, affect performance? (4) Can the learned vocabulary and model attention provide interpretable insights into graph structures?

### 4.1 EXPERIMENTAL SETUP

**Datasets.** We evaluate our method on 14 diverse public datasets for graph classification and regression. The benchmarks span multiple domains, including molecular graphs such as Mutagenicity (`Muta`) and Proteins (Riesen & Bunke, 2008), OGBG-molhiv (Hu et al., 2020), ZINC (Irwin et al., 2012), AQSOL (Sorkun et al., 2019), and QM9 (Wu et al., 2018); computer vision graphs like COIL-DEL (Rossi & Ahmed, 2015); graph theory like Colors-3 (Knyazev et al., 2019) and Synthetic (Feragen et al., 2013); biomedical graphs like DD (Dobson & Doig, 2003) (Bechler-Speicher et al., 2024) and Peptides (Freitas et al., 2020); social networks (Twitter (Pan et al., 2015)); and academic networks (DBLP (Pan et al., 2013)). A summary of dataset statistics is provided in Appendix C.1.

**Baselines.** Our approach is benchmarked against a comprehensive set of baselines, ranging from classic GNNs (GCN (Kipf, 2016), GIN (Chen et al., 2019)) to state-of-the-art models, including the powerful GCN+ (Luo et al., 2025), Graph Transformers like GraphGPS (Rampášek et al., 2022), and the serialization-based GraphMamba (Wang et al., 2024). To ensure a fair comparison, all baseline results are from official implementations run on our unified data splits and preprocessing pipeline (Dwivedi et al., 2023; Luo et al., 2025; 2023; Bechler-Speicher et al., 2024). Results on key benchmarks are in the main text; the rest are in Appendix D.

**Implementation Details.** Our proposed method, `GraphTokenizer` (GT), encodes graphs into token sequences that are subsequently processed by a standard Transformer model for downstream tasks. We report results with two Transformer backbones: (1) **GT+BERT**, which adopts the BERT-small architecture (Devlin et al., 2019), and (2) **GT+GTE**, which uses the more recent GTE model with a parameter count comparable to BERT-base (Zhang et al., 2024). By default, the tokenizer applies the Frequency-Guided Eulerian circuit (`Feuler`) serialization method followed by Byte Pair Encoding (BPE) on the resulting sequences. In the main results table, we report the best-performing serialization for each dataset; a comprehensive comparison across all serialization methods is provided in the ablation study (Table 3) and Appendix D.3. Further details on model architectures, dataset splits, and hyperparameter settings are provided in Appendix C.3.

### 4.2 PERFORMANCE RESULTS

We present the main performance comparison on a representative subset of classification and regression benchmarks in Tables 2. For each dataset, we report the mean and standard deviation of the primary evaluation metric on five independent runs.

**Sequence length and efficiency.** Figure 2 illustrates the impact of our tokenizer on efficiency. As shown in Figure 2a, BPE achieves a high compression ratio, reducing sequence lengths from reversible methods to approximately 10% of their original size. Notably, the frequency-guided Eulerian method (`Feuler`) produces more compact sequences post-BPE than its unguided counterpart, confirming that our structure-guided serialization is particularly well-suited for BPE. This compression translates directly to improved training efficiency. Figure 2b shows that with BPE, our approach using a standard Transformer backbone becomes significantly more efficient than specialized Graph Transformers like GraphGPS and even surpasses classic GNNs such as GatedGCN. While the speedup (e.g., $\sim$2.5$\times$ on `zinc` for a 10$\times$ compression) is not linear due to model overhead, the gains are substantial. This demonstrates that our graph tokenization framework not only enables standard sequence models to process graphs but also makes them a highly efficient and performant option for graph learning tasks.

**Classification and Regression.** Table 2 presents the performance on classification and regression benchmarks, reporting the mean and standard deviation over five independent runs. Our approach, particularly with the GTE backbone (GT+GTE), achieves state-of-the-art results on a majority of the 14 benchmarks. On the `ogbg-molhiv` benchmark, for instance, GT+GTE attains an ROC-AUC of $0.876$ on our test split (val $0.903$), significantly exceeding reported leaderboard results (test $0.8475$, val $0.8275$). This strong performance is achieved using an off-the-shelf sequence model without any graph-specific architectural modifications. Furthermore, the framework's effectiveness is evident even with the compact GT+BERT model, which already outperforms strong baselines on several datasets. Critically, scaling up to the larger GT+GTE backbone yields consistent performance gains across the board, demonstrating a clear advantage over many GNN architectures that can suffer from performance degradation with increased model capacity due to issues like over-smoothing. We further validate these findings through an expanded evaluation in Appendix D.2, which includes comparisons against additional state-of-the-art architectures (e.g., Graphormer, FragNet) and LLM-based Graph Foundation Models (e.g., GraphGPT, LLAGA).

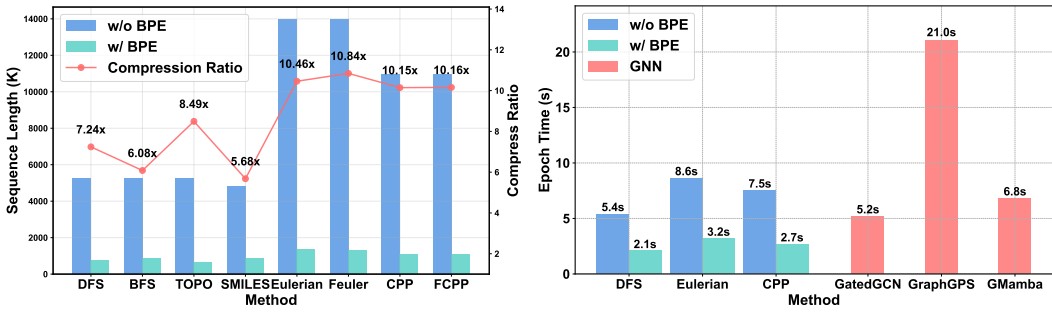

(a) BPE achieves high compression ratios.

(b) Training time per epoch.

Figure 2: Efficiency analysis on the ZINC dataset. (a) BPE greatly reduces token sequence length from serialization. (b) Graph tokenization leads to substantial training speedup by enabling efficient processing with standard Transformers.

Table 2: Results of classification (left block) and regression (right block). The best scores are shown in bold, the second-best are underlined, and standard deviations are given in parentheses. Results for the remaining datasets are presented in Appendix D.

| Model | molhiv auc↑ | p-func ap↑ | mutag acc↑ | coildel acc↑ | dblp acc↑ | qm9 mae↓ | zinc mae↓ | aqsol mae↓ | p-struct avg mae↓ |
|---|---|---|---|---|---|---|---|---|---|
| GCN | 74.0 (0.9) | 53.2 (1.4) | 79.7 (1.7) | 74.6 (0.4) | 76.6 (0.8) | 0.134 (0.004) | 0.399 (0.006) | 1.345 (0.013) | 0.342 (0.003) |
| GIN | 76.1 (1.1) | 61.4 (0.7) | 80.4 (1.2) | 72.0 (0.8) | 73.8 (0.9) | 0.176 (0.006) | 0.379 (0.007) | 2.053 (0.058) | 0.338 (0.002) |
| GAT | 72.1 (0.8) | 51.2 (1.1) | 80.1 (0.9) | 74.4 (1.1) | 76.3 (0.7) | 0.114 (0.015) | 0.445 (0.015) | 1.388 (0.008) | 0.316 (0.003) |
| GatedGCN | 80.6 (0.6) | 51.2 (1.0) | 83.6 (0.8) | 83.7 (0.4) | 86.0 (0.4) | 0.096 (0.007) | 0.370 (0.011) | 0.940 (0.016) | 0.312 (0.004) |
| GraphGPS | 78.5 (1.5) | 53.5 (0.7) | 84.3 (0.9) | 80.5 (0.8) | 71.6 (0.8) | 0.084 (0.004) | 0.310 (0.005) | 1.587 (0.011) | 0.251 (0.001) |
| Exphormer | 82.3 (0.0) | 64.5 (0.9) | 82.7 (1.1) | **91.5** (0.2) | 84.9 (0.4) | 0.080 (0.005) | 0.281 (0.006) | 0.749 (0.005) | 0.251 (0.002) |
| GraphMamba | 81.2 (0.5) | 67.7 (0.9) | 85.0 (1.0) | 74.5 (1.1) | 87.6 (0.5) | 0.083 (0.005) | 0.209 (0.009) | 1.133 (0.014) | 0.248 (0.002) |
| GCN+ | 80.1 (0.6) | 72.6 (0.6) | 88.7 (0.6) | 88.9 (0.3) | 89.6 (0.4) | 0.077 (0.003) | **0.116** (0.009) | 0.712 (0.009) | 0.244 (0.001) |
| **GT+BERT** | 82.6 (0.4) | 68.5 (0.5) | 87.5 (0.9) | 74.1 (0.4) | 93.2 (0.1) | 0.122 (0.008) | 0.241 (0.011) | 0.648 (0.008) | 0.247 (0.002) |
| **GT+GTE** | **87.4** (0.2) | **73.1** (0.2) | **90.1** (0.7) | 89.6 (0.2) | **93.6** (0.1) | **0.071** (0.004) | 0.131 (0.007) | **0.609** (0.016) | **0.242** (0.001) |

## 4.3 ABLATION STUDIES

We conduct ablation studies to evaluate the impact of different serialization methods and the BPE tokenization step while keeping the GT+GTE backbone fixed. Table 3 shows that the choice of serialization has a significant impact on performance. Reversible methods that traverse every edge (e.g., Eulerian and CPP variants) significantly outperform non-reversible node-list traversals, with only a few exceptions detailed in Appendix D. Within the reversible category, the frequency-guided Eulerian circuit (`Feuler`) demonstrates a clear advantage over its unguided counterpart, not only in mean performance but also in reduced variance, indicating greater stability. In contrast, the performance gap between CPP and its frequency-guided version (`FCPP`) remains minimal. A plausible explanation is that CPP's objective of finding a minimum-weight traversal already yields a highly structured sequence, leaving limited room for further improvement from frequency-based guidance. Although `FCPP` performs comparably to `Feuler`, `Feuler` provides substantial benefits in algorithmic complexity (Appendix B.5) and scalability, making it a more practical choice for larger graphs.

A second key finding is that applying BPE to serialized sequences substantially improves model performance. Across nearly all configurations, BPE yields higher scores with a clear performance margin. This improvement is accompanied by reduced standard deviation, indicating more stable and reliable training. Moreover, these performance gains come in addition to the substantial efficiency improvements discussed previously in Figure 2. Therefore, BPE is a critical component of our framework, enhancing both accuracy and computational efficiency.

Additional ablation studies in Appendix D.3 further investigate the sensitivity to the BPE vocabulary size $K$, the choice of statistical guidance unit for frequency collection, and the impact of serialization strategies on graph counting tasks (e.g., the COLORS-3 dataset).

**Visualizing the Learned Vocabulary.** To examine the structural semantics captured by our tokenizer, we visualize the vocabulary constructed by BPE on the ZINC dataset. As shown in Figure 3, BPE iteratively merges simple, frequent substructures into progressively more complex and chem-

Table 3: Ablation of serialization method orderings with and without BPE. The best scores are shown in bold, the second-best are underlined, and standard deviations are given in parentheses. A dash ("—") under the SMILES method indicates that the dataset either lacks SMILES representations or does not correspond to a molecular graph.

| Method | molhiv auc↑ | | coildel acc↑ | | p-func ap↑ | | zinc mae↓ | | qm9 mae↓ | |
|---|---|---|---|---|---|---|---|---|---|---|
| | w | w/o | w | w/o | w | w/o | w | w/o | w | w/o |
| BFS | 72.3 (0.6) | 81.2 (0.9) | 81.2 (0.9) | 80.1 (1.3) | 68.5 (0.6) | 67.2 (0.2) | 0.453 (0.011) | 0.696 (0.013) | 0.311 (0.009) | 0.292 (0.011) |
| DFS | 76.0 (0.4) | 79.1 (0.5) | 80.5 (0.4) | 79.8 (0.8) | 71.0 (1.1) | 68.4 (0.3) | 0.446 (0.009) | 0.705 (0.008) | 0.291 (0.007) | 0.277 (0.010) |
| TOPO | 73.2 (0.6) | 75.6 (0.8) | 82.6 (0.8) | 81.4 (1.2) | 67.9 (0.3) | 64.5 (0.5) | 0.416 (0.010) | 0.634 (0.011) | 0.293 (0.010) | 0.275 (0.013) |
| Eulerian | 84.5 (0.7) | 81.0 (1.0) | 84.1 (1.5) | 84.0 (1.5) | 69.1 (0.6) | 66.8 (1.1) | 0.164 (0.009) | 0.160 (0.016) | 0.083 (0.004) | 0.104 (0.008) |
| Feuler | 87.4 (0.4) | 81.3 (0.5) | 88.0 (0.7) | 85.6 (0.6) | 73.1 (0.3) | 68.1 (0.9) | 0.131 (0.007) | 0.171 (0.013) | 0.071 (0.005) | 0.088 (0.007) |
| CPP | 86.9 (0.3) | 81.2 (0.5) | 89.6 (0.1) | 86.7 (0.3) | 69.2 (0.2) | 67.0 (0.8) | 0.141 (0.006) | 0.145 (0.009) | 0.073 (0.004) | 0.093 (0.006) |
| FCPP | 86.4 (0.3) | 81.0 (0.6) | 89.4 (0.3) | 86.8 (1.0) | 69.2 (0.3) | 66.3 (0.5) | 0.140 (0.005) | 0.151 (0.008) | 0.079 (0.005) | 0.095 (0.007) |
| SMILES | — | — | — | — | — | — | 0.201 (0.012) | 0.339 (0.009) | 0.092 (0.008) | 0.081 (0.014) |

ically meaningful tokens. For example, a basic sulfonyl group is first established as a token, and then combined with adjacent atoms to form larger functional groups. This confirms that BPE can automatically discover and compose semantically relevant substructures, yielding an interpretable, hierarchical vocabulary.

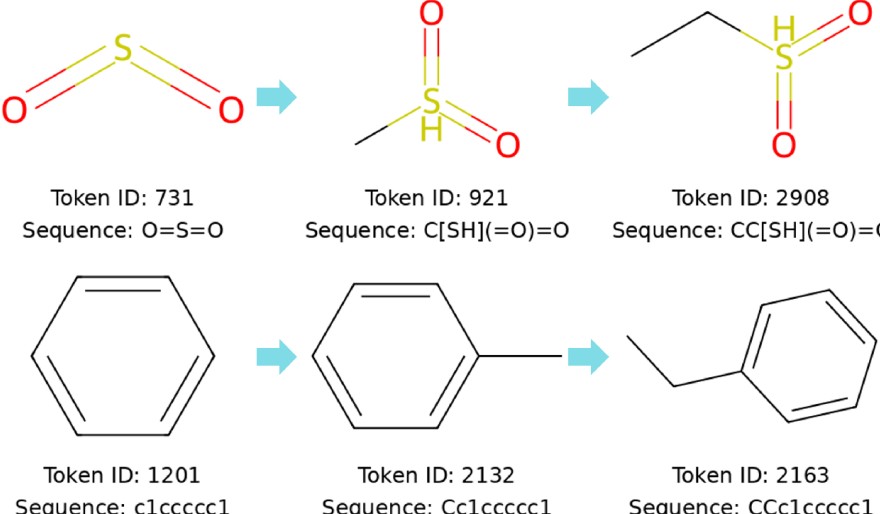

Figure 3: Illustration of the BPE merging process on ZINC. Each row shows how simple substructures (left) are iteratively merged to form larger, chemically meaningful tokens (middle and right).

A statistical analysis of the vocabulary ($K{=}2000$) on ZINC further supports these observations: the token size distribution peaks in the 4–6 node range (41.5%), which corresponds to typical functional groups, while atomic tokens (0–1 nodes) comprise only 7.1%. Over 60% of the vocabulary represents multi-node substructures, confirming that BPE discovers an optimal level of abstraction. Further analysis is provided in Appendix D.1.

## 5 CONCLUSION

In this paper, we introduce a general framework for graph tokenization that bridges graph-structured data with the Transformer ecosystem. Our approach combines reversible, structure-guided graph serialization with BPE to construct a faithful and efficient interface that encodes graphs into discrete token sequences. Fundamentally, the framework decouples the design of graph representations from the underlying model architecture, thereby reframing graph learning as a sequence modeling problem. This perspective enables graph data to be directly integrated into general-purpose Transformers, allowing the graph learning field to leverage rapid advancements in model architectures, training strategies, and scaling capabilities. Empirically, we demonstrate that this approach enables standard off-the-shelf Transformers to process graph data effectively and achieve state-of-the-art results on a diverse set of benchmarks, outperforming established GNNs and specialized Graph Transformers. Limitations of this work and directions for future research are discussed in Appendix A.

## ACKNOWLEDGMENTS

This work is supported in part by the National Natural Science Foundation of China (No. 62550138, 62192784, 62572064, 62472329), and Young Elite Scientists Sponsorship Program (No.2023QNRC001) by CAST.

## ETHICS STATEMENT

The authors of this work have read and commit to adhering to the Code of Ethics. Our research proposes a foundational framework for graph tokenization and, to the best of our knowledge, does not present any direct ethical concerns. The work does not involve the use of personally identifiable information, sensitive human-subject data, or applications with immediate potential for societal harm.

## REPRODUCIBILITY STATEMENT

To ensure full reproducibility, our complete source code is provided in the supplementary materials. This source code contains training configuration files for all experiments, and the necessary scripts to preprocess datasets from their original sources. For convenience, ready-to-use versions of the datasets are also provided. Comprehensive details on the experimental setup are documented in the Appendix, including the datasets (Appendix C.1), model architectures (Appendix C.2), hyperparameter configuration (Appendix C.3), and runtime environment (Appendix C.4).

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

# Appendix

## A  DISCUSSION

This section provides a candid discussion of our framework's limitations and outlines promising avenues for future research.

### A.1  LIMITATIONS

**Graphs with Continuous Features.**  Our framework operates by first converting a labeled graph into a sequence of discrete symbols, which subsequently forms the input for Byte Pair Encoding (BPE). This process implicitly assumes that the features associated with nodes and edges are discrete and can be directly mapped to an initial symbol alphabet. Consequently, graphs where attributes are primarily continuous (e.g., real-valued vectors) cannot be handled natively. The required conversion of continuous attributes into a finite set of discrete symbols always needs a quantization step, which is inherently lossy. This directly conflicts with our framework's core principle of creating a faithful and reversible representation. Integrating continuous features in a principled manner alongside our discrete tokenization process therefore remains a key limitation. However, as elaborated in Appendix A.2, this is not a fundamental blockage. We propose that continuous features can be effectively integrated via Vector Quantization (VQ) interfaces or by treating continuous attributes as a parallel modality fused with token embeddings.

**Node and Edge Level Tasks.**  The current framework is primarily evaluated on graph-level prediction tasks. Adapting it for fine-grained objectives, such as node classification or link prediction, raises a general consideration for graph tokenization: the Byte Pair Encoding (BPE) process, which is essential for building an efficient vocabulary, may merge the specific target node or edge into a larger, composite token. As a result, the discrete identity of the target entity can be obscured, making it difficult to formulate a direct prediction objective. To mitigate this, future iterations can introduce special "pointer tokens" to distinguish original nodes in a composite token or modify the BPE merge rules to prevent target nodes from being merged into composite tokens, thereby preserving their distinct addressability.

**Computational Complexity and Scalability.**  A key scalability limitation in our framework arises from the trade-off between different serialization methods. The Chinese Postman Problem (CPP) based approach, for instance, introduces a significant bottleneck: its $O(|\mathcal{V}|^3)$ complexity renders it impractical for datasets with large graphs. Consequently, we primarily adopt the highly efficient, linear-time `Feuler` method. With this choice, the main computational workload shifts to the BPE vocabulary training. However, this is a far more manageable constraint, as BPE training is a one-time, offline cost, and its practical runtime impact is a constant-factor consideration rather than a prohibitive asymptotic scaling issue (as detailed in Appendix B.5). A separate, downstream constraint is the Transformer's fixed context window, which limits the maximum size of a single graph. While this is an inherent limitation of the model architecture, our BPE compression significantly alleviates the issue, and adopting long-context models provides a clear path for future scaling.

### A.2  FUTURE WORKS

Our work opens several promising avenues for future research. Below, we outline these potential directions, ordered based on our perspective of their potential impact and research scope.

- **Graphs with Continuous Features.** To extend our framework to graphs with continuous features, a straightforward approach would be to first discretize them using methods like vector quantization (Yang et al., 2023) and then apply our pipeline. However, we argue this direction is suboptimal, as such quantization is inherently lossy and conflicts with our framework's core principle of creating a faithful, reversible representation. A more promising direction is to treat discrete and continuous information in parallel channels. For instance, features learned via message passing could be incorporated as a continuous bias added to the discrete token embeddings, analogous to how positional encodings supplement token information in standard Transformers. Alternatively, spectral graph theory offers a principled way to derive global information; the eigenvectors of the graph Laplacian could

be used to generate a unique continuous encoding for each token's position within the global structure, complementing the discrete sequence.

- **Node and Edge Level Tasks.** For adapting the framework to node or edge-level tasks, an intuitive strategy might be to predict the entire composite token that contains the target entity. However, this approach is problematic as it makes the learning signal less direct and can amplify prediction errors. A more targeted and principled approach would be to modify the BPE procedure itself. Specifically, we propose fixing the target entities so they are excluded from merging during tokenization. This would preserve the target's granularity for direct prediction, introducing an interesting trade-off between task-specific fidelity and overall vocabulary efficiency that warrants further investigation.

- **Generative Pre-training and Cross-Domain Generalization.** A key advantage of our sequence-based representation is its potential for large-scale pre-training. A natural next step is to explore pre-training on large corpora of graphs from within the same domain to enhance transferability. More ambitiously, our framework offers a new perspective on the grand challenge of cross-domain generalization. Since our method transforms any graph into a standard sequential format, we hypothesize that graphs from disparate domains can be treated as different 'languages' in NLP. **This analogy extends to generative modeling: just as LLMs generate text, our framework enables autoregressive graph generation. We provide a preliminary verification of this capability in Appendix D.1, demonstrating that a standard decoder-only Transformer can generate valid graph structures.** Consequently, training a single, large-scale Transformer on a diverse, multi-domain corpus of tokenized graphs could facilitate unprecedented knowledge transfer, potentially giving rise to the scaling laws and unified representations that are foundational to true Graph Foundation Models. Finally, such pre-training may also serve as a novel tool for analyzing datasets. Our preliminary results suggest that a dataset's size does not always correlate with its 'information density,' and monitoring a model's overfitting on a masked prediction task could provide a new way to quantify a graph dataset's quality and diversity.

- **Further Extensions and Applications.** Our framework also opens several avenues for direct applications and algorithmic refinements. By reframing graph learning as a sequence modeling problem, it immediately enables the use of powerful autoregressive models, such as GPT-style architectures, for graph generation tasks like controllable molecular design. Furthermore, the sequential representation allows our method to seamlessly integrate with emerging long-context architectures to enhance scalability for massive graphs. Extending the serialization to richer graph structures, such as heterogeneous information networks (Yan et al., 2024) and hypergraphs (Yan et al., 2023), is another promising direction. Finally, a more fundamental extension could involve making the serialization process itself learnable, where traversal decisions are optimized end-to-end for a specific downstream task. While this could create a highly specialized graph-to-sequence interface, it also introduces significant challenges regarding generalization and stability.

## A.3 Use of Large Language Models

In this work, we used large language models (LLMs) to assist with two non-substantive aspects of the research workflow. No part of the scientific contributions—such as algorithm design, model architecture, theoretical formulation, or experimental evaluation—was generated by or delegated to an LLM.

- **Manuscript Editing.** LLMs were used to help polish the language of the manuscript. This includes surface-level edits such as improving clarity, grammar, and conciseness of English expressions. All technical content, algorithmic designs, and empirical results were authored and validated by the authors.

- **Code Documentation and Cleanup.** At the time of open-sourcing our implementation, LLMs were used to assist in non-algorithmic tasks including: adding docstrings and in-line comments, generating basic usage documentation, removing deprecated or redundant code, and improving logging output for better reproducibility. All code functionality and correctness were manually verified by the authors.

# B    FURTHER METHODOLOGICAL DETAILS

This section provides a formal and detailed supplement to the methodological discussions in Section 3. We aim to precisely define how classical graph traversal algorithms are adapted for the task of graph serialization, and to rigorously analyze their resulting properties of **reversibility** and **determinism**, pinpointing the exact sources of their respective strengths and weaknesses for our framework. All notations, unless specified otherwise, follow the definitions established in Section 3.1.

## B.1    DISTINCTION FROM NEURAL GRAPH TOKENIZATION METHODS

It is crucial to clarify that BPE differs fundamentally from **neural-based continuous encoders** such as VAE-based subgraph tokenizers (e.g., PS-VAE (Kong et al., 2022)), linear projectors, or ViT-style patchified embeddings. These methods utilize learnable parameters to project discrete inputs into **continuous embedding spaces**. While such continuous approximation is native for processing continuous signals (e.g., pixel intensities in CV), it introduces inevitable information loss and reconstruction ambiguity when applied to discrete structures like graphs. In contrast, our BPE approach is a deterministic, statistics-based algorithm operating entirely within the discrete symbolic domain. It does not require neural network training and merges symbols based on frequency, avoiding the gradient-based optimization complexity inherent in forcing discrete topology into continuous representations.

This distinction yields decisive practical advantages: (1) **Reversibility:** Our framework is inherently lossless and deterministic. Since graph serialization preserves topology and BPE merging is reversible via the vocabulary, we can reconstruct the exact graph structure up to isomorphism, whereas VAEs rely on probabilistic decoders that only approximate the input. (2) **Efficiency:** BPE achieves significant sequence compression (over $10\times$) without the computational overhead of training separate neural encoders, effectively reducing the quadratic complexity of the Transformer's attention mechanism. (3) **Decoupling:** By converting graphs into standard discrete symbol sequences, we completely decouple graph representation from model architecture. Neural encoding strategies typically require specific graph encoders (e.g., GNNs) or specialized projection layers to bridge the modality gap. Once tokenized by our framework, a graph is formally identical to a natural language sequence and can be processed by any unmodified, off-the-shelf Transformer without specialized graph encoders or projection layers.

## B.2    FORMAL ANALYSIS OF NODE-LIST SERIALIZATION METHODS

We first consider the class of methods that generate a sequence composed solely of node identifiers.

**Formal Definition.**    When applied to graph serialization, a node-list traversal method is a function $f_{\text{node}} : \mathcal{G} \mapsto S$ that maps a labeled graph $\mathcal{G} = ((\mathcal{V}, \mathcal{E}), L, \Sigma)$ to a sequence of node labels $S = (s_1, s_2, \ldots, s_{|\mathcal{V}|})$. This sequence is generated by a traversal that visits every node in $\mathcal{V}$ exactly once. Let the sequence of visited nodes be $(v_1, v_2, \ldots, v_{|\mathcal{V}|})$, where $\{v_1, \ldots, v_{|\mathcal{V}|}\} = \mathcal{V}$. The output sequence is then $S = (L(v_1), L(v_2), \ldots, L(v_{|\mathcal{V}|}))$.

- For **Breadth-First Search (BFS)** and **Topological Sort**, the order of nodes in the sequence reflects the layer-by-layer or dependency-based traversal order. It is crucial to note that for $i \in [1, |\mathcal{V}| - 1]$, an edge $(v_i, v_{i+1})$ is not guaranteed to exist in $\mathcal{E}$.
- For **Depth-First Search (DFS)**, the standard algorithm produces a sequence where each node $v_{i+1}$ is an unvisited neighbor of $v_i$ (or a node reached after backtracking). The sequence of *first discovery* implies a tree structure (the DFS tree), but the output sequence itself does not encode this structure explicitly.

**Reversibility.**    These methods are fundamentally **irreversible**. A serialization function $f$ is reversible if, for its output $S = f(\mathcal{G})$, the set of pre-images $f^{-1}(S)$ contains a graph $\mathcal{G}'$ such that $\mathcal{G}' \cong \mathcal{G}$. For node-list methods, this condition fails. The output sequence $S$ discards all explicit edge connectivity information. Consequently, a vast number of non-isomorphic graphs can produce the exact same node-label sequence. For example, consider two graphs on three nodes $\{A, B, C\}$ with identical labels: a path graph $\mathcal{G}_1$ with edges $\{(A, B), (B, C)\}$ and a star graph $\mathcal{G}_2$ with edges $\{(A, B), (A, C)\}$. A valid BFS starting from node A in $\mathcal{G}_1$ could produce the sequence

$(L(A), L(B), L(C))$, and a BFS starting from A in $\mathcal{G}_2$ could also produce the same sequence. From the sequence alone, it is impossible to distinguish $\mathcal{G}_1$ from $\mathcal{G}_2$, thus violating reversibility.

**Determinism.** These methods are inherently **non-deterministic**. The source of non-determinism lies in the arbitrary selection of the next node to visit from a set of valid candidates. Formally, at any step of the traversal from a node $u$, let $N_{\text{valid}}(u)$ be the set of unvisited neighbors (for BFS/DFS) or nodes with no remaining incoming edges (for Topological Sort). Since $N_{\text{valid}}(u)$ is an unordered set, any choice of $v \in N_{\text{valid}}(u)$ is permissible by the standard algorithm's definition. The final sequence is thus contingent on implementation-specific details, such as the memory layout of the graph's adjacency list, which are not canonical properties of the graph itself. This leads to different sequences for the same graph, violating determinism.

### B.3 FORMAL ANALYSIS OF EDGE-COVERING SERIALIZATION METHODS

This class of methods generates sequences by performing a walk that traverses every edge in the graph at least once.

**Formal Definition.** A *walk* $W$ in a graph $\mathcal{G}$ is a finite sequence of alternating nodes and edges, $W = (v_0, e_1, v_1, e_2, \ldots, e_k, v_k)$, where $v_i \in \mathcal{V}$, $e_i \in \mathcal{E}$, and for all $i \in [1, k]$, edge $e_i = (v_{i-1}, v_i) \in \mathcal{E}$. An edge-covering serialization function $f_{\text{edge}} : \mathcal{G} \mapsto S$ maps $\mathcal{G}$ to the label sequence corresponding to such a walk, $S = (L(v_0), L(e_1), L(v_1), \ldots, L(v_k))$, under the constraint that the multiset of edges in the walk, $\{e_1, \ldots, e_k\}$, covers the entire edge set of the graph, i.e., $\mathcal{E} \subseteq \{e_1, \ldots, e_k\}$.

- For an **Eulerian circuit**, the algorithm is applied to a graph where every undirected edge $\{u, v\} \in \mathcal{E}$ is treated as a pair of directed edges, $(u, v)$ and $(v, u)$, forming an edge set $\mathcal{E}'$ (Gao et al., 2025). The resulting walk traverses every edge in $\mathcal{E}'$ *exactly once*.

- For the **Chinese Postman Problem (CPP)**, the algorithm finds a walk $W$ that covers $\mathcal{E}$ while minimizing the total weight of the walk, $\sum_{i=1}^{k} w(e_i)$, where $w(e)$ is the weight of edge $e$ (typically 1 for unweighted graphs). This means some edges in $\mathcal{E}$ may be traversed multiple times.

**Reversibility.** By definition, these methods are **reversible**. The output sequence $S$ is composed of a series of labeled triplets $(L(v_{i-1}), L(e_i), L(v_i))$. From this sequence, one can reconstruct the complete multiset of labeled edge traversals. Since this multiset is guaranteed to contain every edge from the original graph $\mathcal{E}$ at least once, the full topology of $\mathcal{G}$ can be losslessly recovered up to isomorphism. The mapping from the graph's structure to the information contained in the sequence is injective.

**Determinism.** In their classical forms, these methods are **non-deterministic**.

- For an **Eulerian Path**, the non-determinism stems from Hierholzer's algorithm. At any node $u$ during the tour construction, let $E_u^{\text{unvisited}}$ be the set of untraversed edges incident to $u$. The algorithm proceeds by selecting an arbitrary edge from this set. As $E_u^{\text{unvisited}}$ is an unordered set, this choice introduces ambiguity, leading to different valid Eulerian circuits and thus different output sequences.

- For the **CPP**, the primary source of non-determinism arises during the solution process. The algorithm first identifies the set of odd-degree nodes, $\mathcal{V}_{\text{odd}}$. It then computes a minimum-weight perfect matching on a complete auxiliary graph constructed on $\mathcal{V}_{\text{odd}}$. If multiple distinct perfect matchings exist that share the same minimum total weight, the choice between them is arbitrary. This choice dictates which paths in the original graph are duplicated to form an Eulerian supergraph, ultimately resulting in different minimum-cost tours and thus different sequences.

Our proposed frequency-guided mechanism (Section 3.2) addresses these specific sources of non-determinism by providing a canonical, data-driven rule for making these choices.

### B.4 DISCONNECTED GRAPHS

For completeness, we specify our procedure for handling graphs that are not connected. To ensure the overall serialization remains deterministic, an input graph $\mathcal{G}$ is first decomposed into its set of

connected components, $\{\mathcal{G}_1, \mathcal{G}_2, \ldots, \mathcal{G}_c\}$. Each component $\mathcal{G}_i$ is independently serialized into a sequence $S_i$ using the chosen method. The resulting set of sequences $\{S_1, \ldots, S_c\}$ is then sorted to produce a canonical ordering. The sorting criterion is primarily the length of the sequence in descending order, any ties are resolved using standard lexicographical comparison. The final output is the concatenation of these sorted sequences. This guarantees that any given graph, regardless of its connectivity, maps to a single, unique sequence.

### B.5 COMPLEXITY ANALYSIS

**CPP Complexity.** For CPP–based serialization, the end-to-end cost is $O(|\mathcal{V}|^3 + |\mathcal{E}|)$: $O(|\mathcal{V}|^3)$ from the minimum-weight perfect matching on odd-degree vertices, and $O(|\mathcal{E}|)$ from finding the Euler circuit after augmentation. Since $|\mathcal{E}| \leq |\mathcal{V}|^2$ (even $|\mathcal{E}| = \Theta(|\mathcal{V}|^2)$ for a complete graph), we have $O(|\mathcal{V}|^3 + |\mathcal{E}|) = O(|\mathcal{V}|^3)$, so throughout we denote the CPP family as $O(|\mathcal{V}|^3)$.

**Pipeline Complexity.** We analyze the computational complexity of the `GraphTokenizer` training procedure. Let $\mathcal{D} = \{\mathcal{G}_1, \ldots, \mathcal{G}_N\}$ be the training dataset, where $\mathcal{G}_i = (\mathcal{V}_i, \mathcal{E}_i)$. We define $V_S = \sum_{i=1}^{N} |\mathcal{V}_i|$ and $E_S = \sum_{i=1}^{N} |\mathcal{E}_i|$ as the total number of nodes and edges in the dataset, respectively. Let $K$ be the number of BPE merge operations and $\bar{L}$ be the average initial sequence length, where $\bar{L} \approx E_S/N$.

*Statistics Collection.* This stage requires a single pass over all edges in the dataset to count local patterns. The complexity is therefore $O(E_S)$.

*Graph Serialization.* As summarized in Table 1, the complexity depends on the chosen method. Traversal-based methods such as `Eulerian` (and its guided variant `Feuler`) are linear in graph size, with a total cost for the dataset of $O(V_S + E_S)$. In contrast, CPP (and its guided variant `FCPP`) is dominated by solving a minimum-weight perfect matching for each graph, yielding a total complexity of $O(\sum_{i=1}^{N} |\mathcal{V}_i|^3)$. Assuming a relatively homogeneous distribution of graph sizes, this can be expressed as $O(N \cdot (\frac{V_S}{N})^3) = O(\frac{V_S^3}{N^2})$.

*BPE Training.* The complexity of the BPE training phase (Lines 6-14) is dominated by the sequence merge operation. For a standard implementation using dynamic arrays, the cost of a merge is proportional to sequence length. We analyze the total complexity under two bounding scenarios.

First, consider the case where a small number of pairs are merged per iteration. The work is dominated by rewriting all $N$ sequences for each of the $K$ iterations, leading to a total complexity of:

$$C_1 = O(K \cdot N \cdot \bar{L}) \tag{8}$$

Second, consider the case where a maximal number of pairs are merged, halving the total corpus length $n = \frac{N\bar{L}}{2}$ in each step. The work performed can be described by the recurrence relation:

$$T(n) = T(\frac{n}{2}) + f(n) \tag{9}$$

where $f(n)$ is the cost of a single merge pass. For a naive array-based merge, $f(n) = O(N \cdot (\frac{n}{N})^2) = O(\frac{n^2}{N})$. By the Master Theorem, with $a = 1, b = 2, d = 2$, we have $d > \log_b a$, so the total complexity is dominated by the root node's work:

$$C_{2,\text{naive}} = O(\frac{n^2}{N}) = O(N\bar{L}^2) \tag{10}$$

If the merge operation were optimized to be $O(1)$ per merged pair (e.g., using a linked list representation or mark new token only at range endpoint to be replaced), then $f(n) = O(n)$. In this case, $d = 1 > \log_2 1$, so the complexity would be:

$$C_{2,\text{opt}} = O(n) = O(N\bar{L}) \tag{11}$$

The final complexity depends on the termination criterion. We employ a fixed number of iterations, $K$, to allow for flexible control over the final vocabulary size and compression ratio across experiments. In typical settings, $K$ (e.g., $10^3 - 10^4$) is much larger than the average sequence length $\bar{L}$ (e.g., $10^2$), making the $C_1$ term the dominant factor. Alternatively, if one were to terminate based on a minimum frequency threshold, the process would resemble the recursive scenario, making $C_2$

the more relevant complexity model. Given our fixed-iteration approach, the overall BPE training complexity is:

$$C_{\text{BPE}} = O(K \cdot N \cdot \bar{L}) = \boldsymbol{O(K \cdot E_S)} \tag{12}$$

**Overall Training Complexity.** For serialization with `Feuler` and other methods with linear time complexity, the total complexity is dominated by the BPE training, resulting in $\boldsymbol{O(K \cdot E_S)}$. If a minimum-frequency stopping criterion were used, the complexity would instead be dominated by the $C_{2,\text{naive}}$ term, becoming $O(E_S + N\bar{L}^2) = O(N\bar{L}^2) = O(\frac{E_S^2}{N})$. For serialization with `FCPP`, the total complexity is dominated by the most expensive component, yielding $\boldsymbol{O(\frac{V_S{}^3}{N^2} + K \cdot E_S)} = \boldsymbol{O(\frac{V_S{}^3}{N^2})}$, which in practice for large graphs, implies that the serialization becomes the bottleneck.

## C  EXPERIMENTAL SETUP

This section provides the comprehensive configuration details required to fully reproduce all experiments presented in this paper.

### C.1  DATASETS

Our evaluation is conducted on a diverse suite of benchmark datasets. The `aqsol` and `zinc` datasets are sourced from the benchmark collection proposed by (Dwivedi et al., 2023), while the remaining datasets are standard benchmarks obtained from libraries such as PyTorch Geometric (PyG) and DGL. Table 4 below offers a comprehensive summary of their statistical properties and raw feature dimensions.

Table 4: Comprehensive statistics and feature dimensions for all benchmark datasets. Node and edge counts are presented as mean $\pm$ standard deviation. "Raw Dim" refers to the dimensionality of the original features before they are mapped to discrete integer symbols.

| Dataset | # Graphs | Task | # Targets | Nodes (Mean $\pm$ Std) | Edges (Mean $\pm$ Std) | Node Raw Dim | Edge Raw Dim |
|---|---|---|---|---|---|---|---|
| aqsol | 9,823 | Regression | 1 | $33.7 \pm 24.5$ | $67.9 \pm 50.0$ | 65 | 5 |
| coildel | 3,900 | Classification | 100 | $21.5 \pm 13.2$ | $108.5 \pm 77.0$ | 2 | 1 |
| colors3 | 10,500 | Classification | 11 | $61.3 \pm 60.5$ | $182.1 \pm 187.3$ | 4 | — |
| dblp | 19,456 | Classification | 2 | $10.5 \pm 8.5$ | $39.3 \pm 39.3$ | 1 | 1 |
| dd | 1,178 | Classification | 2 | $284.3 \pm 272.1$ | $1431.3 \pm 1388.4$ | 1 | — |
| molhiv | 41,127 | Classification | 2 | $25.5 \pm 12.1$ | $54.9 \pm 26.4$ | 9 | 3 |
| muta | 4,337 | Classification | 2 | $30.3 \pm 20.1$ | $61.5 \pm 33.6$ | 1 | 1 |
| p-func | 15,535 | MT-Classification | 10 | $150.9 \pm 84.2$ | $307.3 \pm 172.2$ | 9 | 3 |
| p-struct | 15,535 | MT-Regression | 11 | $150.9 \pm 84.2$ | $307.3 \pm 172.2$ | 9 | 3 |
| proteins | 1,113 | Classification | 2 | $39.1 \pm 45.8$ | $145.6 \pm 169.3$ | 2 | — |
| qm9 | 130,831 | Regression | 16 | $18.0 \pm 2.9$ | $37.3 \pm 6.3$ | 16 | 4 |
| synthetic | 300 | Classification | 2 | $100.0 \pm 0.0$ | $392.0 \pm 0.0$ | 2 | — |
| twitter | 144,033 | Classification | 2 | $4.0 \pm 1.7$ | $10.0 \pm 9.1$ | 1 | 1 |
| zinc | 12,000 | Regression | 1 | $43.8 \pm 8.5$ | $91.1 \pm 18.1$ | 28 | 4 |

### C.2  MODEL ARCHITECTURES

We employed two Transformer backbones in our experiments. The first, which we denote as **GT+BERT**, is based on a BERT-Small architecture. The second, **GT+GTE**, utilizes a more recent and powerful GTE-Base model. The precise architectural parameters for each are detailed side-by-side in Table 5. This format facilitates direct comparison and is designed to accommodate additional model configurations in future scaling law studies. Note that the vocabulary size is determined dynamically based on the dataset and tokenization strategy, causing the total parameter count to vary slightly across experiments, the results reported here include the embedding layer size corresponding to the vocabulary encoded using BPE on the ZINC dataset.

### C.3  HYPERPARAMETERS

Our hyperparameter tuning strategy was designed for systematic evaluation and reproducibility rather than exhaustive per-dataset optimization. We established a robust base configuration, detailed in our publicly available configuration files, which was applied to all experiments by default. For certain datasets, particularly those with very large or very small graphs, targeted adjustments were made to key parameters like batch size and learning rate to ensure stable training.

Table 5: Architectural parameters of the Transformer backbones used in our experiments.

| Parameter | GT+BERT | GT+GTE |
|---|---|---|
| *Model Configuration* | BERT-Small | GTE-Base |
| Number of Hidden Layers ($N$) | 4 | 12 |
| Hidden Size ($d_{\text{model}}$) | 512 | 768 |
| Number of Attention Heads ($h$) | 4 | 12 |
| FFN Intermediate Size ($d_{\text{ff}}$) | 2048 | 3072 |
| Activation Function | GELU | GELU |
| Dropout Rate (Attention & Hidden) | 0.1 | 0.1 |
| Position Embedding | Learned Abs | RoPE |
| Max Sequence Length | 768 | 8192 |
| Layer Normalization $\epsilon$ | 1e-12 | 1e-12 |
| Total Parameters (Approx.) | $\approx$ 15M | $\approx$ 115M |

Table 6 provides a comprehensive overview of these settings. The "Default Configuration" column represents the base values applied to all datasets unless otherwise specified. The subsequent columns detail the specific overrides for datasets or groups of datasets that required adjustments. This unified view clearly illustrates both our general training strategy and the specific exceptions made.

Table 6: Unified hyperparameter specification. This table details the default values used for training and the specific conditions under which they were overridden.

| Parameter | Default Value | Exceptions & Conditions |
|---|---|---|
| *General Settings* | | |
| Optimizer | AdamW | — |
| Epochs | 200 | — |
| Weight Decay | 0.1 | — |
| Random Seed | 42 | — |
| Batch Size | 32 | 16 on (DBLP, `Peptides_*`, and `COIL-DEL`) |
| *Serialization & Tokenizer Settings* | | |
| Default Method | `Feuler` | — |
| BPE Enabled | True | — |
| BPE Merges ($K$) | 2000 | — |
| *Finetuning Settings* | | |
| Learning Rate | 1e-5 | 5e-5 on (DBLP, `molhiv`, `twitter`) |
| LR Warmup Ratio | 0.025 | — |
| Max Gradient Norm | 0.5 | — |
| Early Stopping | 20 epochs | — |
| *Pre-training (MLM) Settings* | | |
| Learning Rate | 1e-4 | 5e-5 on (`muta`, `molhiv`, `qm9`, `twitter`, `dblp`) For GTE, to prevent training instability. |
| LR Warmup Ratio | 0.12 | — |
| Max Gradient Norm | 2.0 | — |
| Mask Probability | 0.09 | — |

## C.4 COMPUTATIONAL ENVIRONMENT

**Software.** Our implementation relies on a shared software stack to ensure consistency. The key packages and their versions are listed below. For a complete and exhaustive list of dependencies, please refer to the environment files in our public code repository.

- **PyTorch**: `2.1.2`
- **CUDA Toolkit**: `12.1`
- **DGL**: `2.4.0`
- **PyTorch Geometric (PyG)**: `2.4.0` (with corresponding libraries `pyg-lib`, `torch-scatter`, etc.)

**Hardware.** All experiments were conducted on a heterogeneous cluster of NVIDIA GPUs, including consumer grade (GeForce RTX 2080, 3090, 4090) and data center grade (A800, H800) hardware. We verified that our results are stable and consistent across these different architectures.

### C.5 HYPERPARAMETER SETTINGS FOR ADDITIONAL BASELINES

To ensure a fair comparison in the extended benchmarks (Appendix D.2), we adopted the official configurations for all baseline models whenever available:

- **Graph Transformers (GraphGT, Graphormer):** We utilized the standard hyperparameters provided in their official implementations (e.g., 6–12 layers, hidden dimension of 80–768 depending on dataset size) to match the parameter budget of our models.
- **Graph Foundation Models (GraphGPT, LLAGA):** For the adaptation experiments, we used the open-sourced checkpoints (e.g., LLaMA-2-7b backbone) and fine-tuned the projection layers or LoRA adapters following the protocols specified in their original papers.
- **Specialized Architectures (FragNet, Graph-ViT-MLPMixer):** We reproduced these results using the optimal settings reported in their respective publications.

## D EXPERIMENTAL RESULTS

In this section, we present a comprehensive suite of additional experiments to further validate the robustness, interpretability, and comparative performance of our proposed framework. The content is organized as follows:

- **Qualitative Analysis (Appendix D.1):** We offer a detailed interpretation of the learned BPE vocabulary to elucidate the structural semantics captured by our tokenizer, and provide a proof-of-concept for autoregressive graph generation.
- **Performance Benchmarks (Appendix D.2):** We report complete results for all datasets and expand our comparative evaluation to include recent state-of-the-art Graph Transformers (e.g., GraphGT, Graphormer) and LLM-based Graph Foundation Models (e.g., GraphGPT, LLAGA).
- **Ablation Studies (Appendix D.3):** We conduct systematic investigations into core design choices, ranging from architectural components (e.g., Transformer backbones, serialization methods) to specific hyperparameters (e.g., vocabulary size, statistical guidance units), and empirically validate the impact of serialization strategies on counting tasks.
- **Scalability Analysis (Appendix D.4):** We report runtime measurements on large-scale OGB datasets to empirically validate the scalability of our pipeline.

### D.1 QUALITATIVE ANALYSIS AND INTERPRETABILITY

**Visualizing the Learned Vocabulary.** To understand the structural patterns captured by our tokenizer, we visualize the vocabulary constructed by BPE's merging process on the ZINC dataset. Figure 3 illustrates how BPE iteratively merges simple, frequent substructures into progressively more complex and chemically meaningful tokens.

Each row in the figure demonstrates such a merging sequence. For instance, the top row shows that a basic structure representing a sulfonyl group (`O=S=O` at the 731st merge iteration) is established as a token. In subsequent merge steps, BPE combines this token with adjacent atoms to form a more complex token (`C[SH](=O)=O`) and then an even larger one (`CC[SH](=O)=O`). This progression directly reflects BPE's mechanism: greedily merging the most frequent adjacent symbol pairs to build a vocabulary. Similarly, the bottom row shows the process starting from a classic benzene

ring, which is then merged with neighboring carbon chains to form tokens corresponding to toluene and ethylbenzene.

**Interpretation of Learned Vocabulary.** To understand the structural semantics captured by our tokenizer, we analyzed the size distribution of the learned vocabulary on the ZINC dataset with a merge count of $K = 2000$. As detailed in Table 7, the vocabulary is not dominated by small atomic tokens. Instead, it exhibits a distinct preference for medium-scale substructures. Specifically, atomic tokens (0-1 nodes) comprise only 7.1% of the vocabulary, while the distribution peaks in the **4-6 node range** (41.5%). In the context of molecular graphs, this size range corresponds to typical functional groups and ring structures. Combined with the 7-9 node range, over 60% of the vocabulary represents complex, multi-node substructures. This confirms that the BPE merge process, guided by our frequency statistics, successfully identifies an optimal level of abstraction—producing tokens large enough to capture meaningful local topology (e.g., cycles, branches) yet frequent enough to ensure generalization across the dataset.

Table 7: Fine-grained distribution of token sizes (node counts) in the learned BPE vocabulary on ZINC ($K = 2000$). The distribution peaks at the 4-6 node range, indicating a preference for functional-group-sized substructures.

| Token Size | Atomic ($0 \sim 1$) | Small ($2 \sim 3$) | **Medium** ($4 \sim 6$) | Large ($7 \sim 9$) | Huge ($10+$) |
|---|---|---|---|---|---|
| **Proportion** | 7.1% | 28.5% | **41.5%** | 20.4% | 2.5% |

**Proof-of-Concept: Auto-regressive Graph Generation.** A core advantage of our framework is that it converts graph data into a format formally identical to natural language, thereby theoretically enabling the use of standard autoregressive (decoder-only) models for graph generation. To empirically validate this capability, we conducted a proof-of-concept experiment using the MNIST dataset, treating images as readily visualizable grid graphs.

**Setup.** We converted each $28 \times 28$ MNIST image into a regular grid graph, where pixels correspond to nodes and are connected to their immediate spatial neighbors (up, down, left, right). These grid graphs were then tokenized using our proposed framework (Frequency-Guided Serialization + BPE) to produce discrete token sequences. We trained a standard, unmodified decoder-only Transformer (GPT-style architecture) on these sequences using the conventional next-token prediction objective ($\mathcal{L}_{\text{CLM}}$).

**Results.** As illustrated in Figure 4, the model successfully learns to generate coherent graph structures token-by-token. The reconstructed graphs clearly depict recognizable digits, demonstrating that the model effectively captures the global topology and local connectivity patterns solely from the serialized token sequence. This result confirms that our framework effectively bridges the gap between graph generation and standard sequence modeling, opening the door for applying large-scale generative pre-training (e.g., GPT) to graph domains such as molecular design and material discovery.

## D.2 PERFORMANCE RESULTS

**Classification and Regression Benchmarks.** We present the complete results for all datasets evaluated in our main experiments. The experimental setup and reporting format are identical to those described in the main text. The results for the remaining datasets (which were not included in the main text due to space constraints) are detailed in Table 8.

**Comparison with State-of-the-Art Architectures.** To strictly validate the effectiveness of our tokenizer, we significantly expanded our comparative evaluation to include a broader range of state-of-the-art architectures. We conducted a comprehensive evaluation against recent Graph Transformers (**GraphGT**, **Graphormer**), serialization-based or hybrid methods (**FragNet**, **Graph-ViT-MLPMixer**), and classic baselines (**HAN**, **ChebNet**). As detailed in Table 9, our method consistently outperforms strong baselines on graph classification (MOLHIV, COIL-DEL) and long-range modeling tasks (Peptide-func). Notably, we surpass specialized architectures like FragNet and Graph-ViT-MLPMixer without requiring their domain-specific inductive biases. On the ZINC regression task, our performance is comparable to standard Graph Transformers (e.g., Graphormer), while classic architectures (ChebNet, HAN) lag significantly behind. This confirms that a general-purpose tokenizer can unlock state-of-the-art performance across diverse graph learning benchmarks using a standard Transformer architecture.

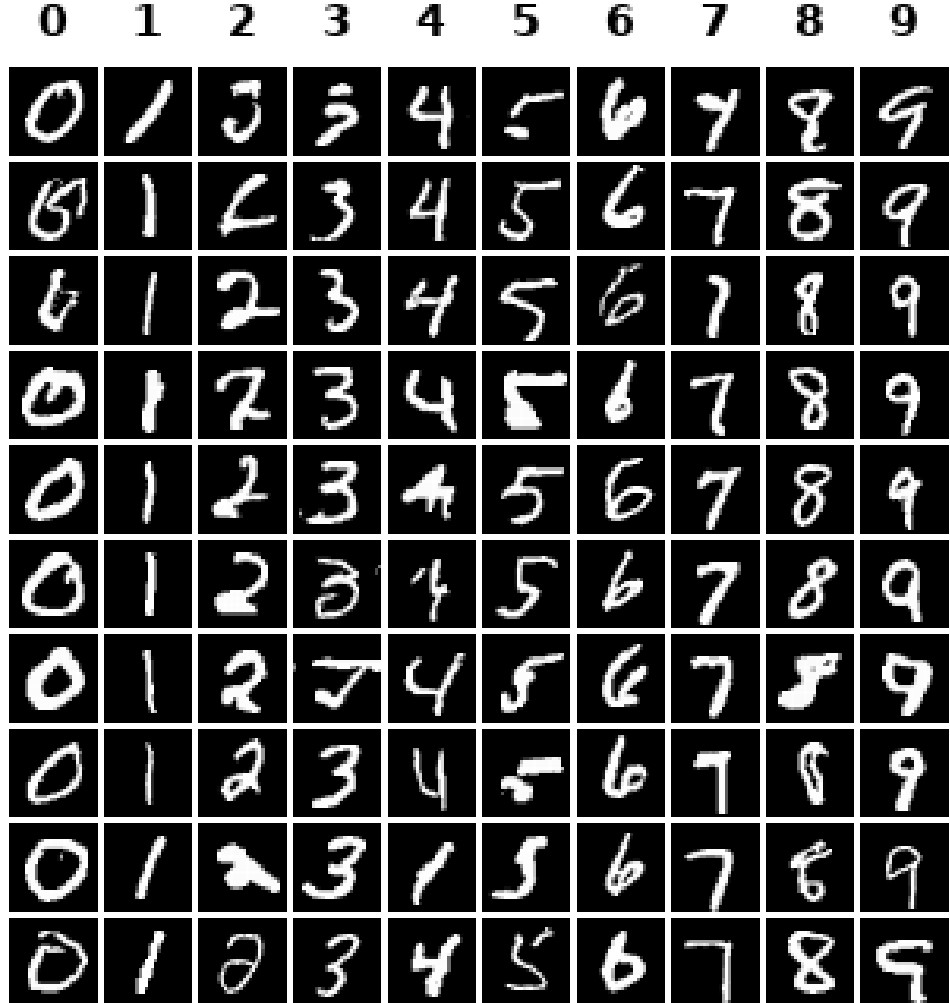

Figure 4: Visualization of autoregressive graph generation on MNIST. The model generates the graph structure token-by-token (left to right). The resulting sequences are decoded back into grid graphs, forming coherent digit images. This demonstrates the framework's capability to support generative tasks using standard decoder-only Transformers.

**Comparison with LLM-based Graph Models.**   We further benchmarked our method against recent "Graph Foundation Models" (GFMs) such as **GraphGPT** and **LLAGA**. Since these models are fundamentally designed for text-attributed graphs (TAGs), we evaluated them via adaptation (textualizing graph structures), treating discrete node labels as textual tokens to satisfy their input constraints. The results in Table 10 reveal a critical limitation of the textualization paradigm. These models struggle significantly with pure structural tasks, performing near random guessing on datasets like COIL-DEL. This highlights that while LLMs excel when rich semantic text is available, they lack the native capability to reason about complex graph topology solely from linearized text strings. In contrast, our tokenizer specifically captures structural patterns, enabling superior performance on structure-centric tasks.

**Sequence length and efficiency.**   To demonstrate the generalizability of our model's efficiency, we present visualizations of token efficiency and training throughput for the remaining datasets, analogous to Figure 2 in the main paper. The results, shown in Figure 5–10, confirm that our approach consistently maintains superior efficiency across a diverse range of datasets.

Table 8: Additional classification results on datasets not shown in the main table. Best scores are in **bold**, second-best are underlined. Std in parentheses.

| Model | colors3 acc↑ | twitter acc↑ | proteins acc↑ | dd acc↑ |
|---|---|---|---|---|
| GCN | 66.0 (2.1) | 52.7 (0.3) | 73.2 (1.6) | 62.7 (1.1) |
| GIN | 69.3 (1.6) | 55.6 (0.2) | 64.3 (4.0) | 65.2 (0.9) |
| GAT | 76.6 (1.4) | 53.6 (0.3) | 70.5 (0.9) | 58.5 (1.0) |
| GatedGCN | 77.1 (1.2) | 59.8 (0.4) | 71.1 (1.2) | 77.2 (0.7) |
| GraphGPS | 77.4 (1.8) | 53.0 (0.5) | 67.9 (0.6) | 76.3 (0.5) |
| Exphormer | 73.9 (1.9) | 55.1 (0.3) | 74.1 (0.9) | 74.6 (0.6) |
| GraphMamba | 93.6 (0.9) | 56.4 (0.7) | 70.5 (1.1) | 76.8 (0.4) |
| GCN+ | 85.8 (2.3) | 61.3 (0.2) | 77.1 (0.9) | 79.1 (0.5) |
| **GT+BERT** | 96.2 (1.7) | 62.3 (0.3) | 75.0 (0.7) | 72.0 (0.9) |
| **GT+GTE** | **100.0** (0.0) | **65.7** (0.2) | **79.1** (0.6) | **79.6** (0.6) |

Table 9: Comparison with additional baselines. '*' indicates results reproduced using official implementations; '–' denotes results not reported in the original papers. Our method remains competitive against specialized architectures.

| Model | ZINC (MAE ↓) | MOLHIV (AUC ↑) | Peptide-func (AP ↑) | COIL-DEL (ACC ↑)* |
|---|---|---|---|---|
| GraphGT | 0.226 (0.014) | – | 63.26 (1.26) | 86.1 (0.8) |
| Graphormer | 0.132 (0.006) | 80.51 (0.53) | – | 88.4 (0.3) |
| FragNet | 0.078 (0.005) | 81.32 (0.86) | 66.8 (0.5) | 83.4 (0.6) |
| Graph-ViT-MLPMixer | **0.073** (0.001) | 79.97 (1.02) | 69.7 (0.8) | 89.1 (0.4) |
| HAN* | 0.348 (0.042) | 72.3 (0.6) | 52.2 (1.1) | 74.5 (0.6) |
| ChebNet* | 0.423 (0.013) | 70.1 (0.7) | 51.3 (0.8) | 71.4 (0.8) |
| **Our Method** | 0.131 (0.007) | **87.4** (0.2) | **73.1** (0.2) | **89.6** (0.2) |

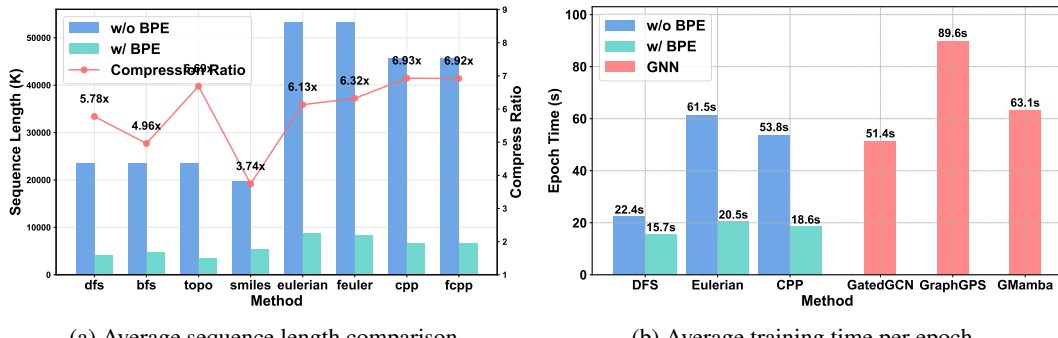

(a) Average sequence length comparison.

(b) Average training time per epoch.

Figure 5: Efficiency analysis on the `QM9` dataset.

### D.3 ABLATION STUDIES

To further validate our design choices, this section provides detailed ablation studies that complement the analysis in the main paper. We first present the complete ablation results for the GT+GTE model in Table 11 and the GT+BERT model in Table 12 and Table 13. Beyond these general performance trends, we conducted systematic studies to validate our specific design choices regarding vocabulary size, frequency guidance units, and serialization strategies for counting tasks.

**Impact of Vocabulary Size** ($K$). Our experiments identify $K = 2000$ as the optimal saturation point, effectively balancing sequence compression with model learnability. As shown in Table 14, increasing $K$ initially yields significant gains in compression (1.00x to 10.84x). However, this benefit plateaus around $K = 2000$. Extending to $K = 5000$ offers diminishing returns in compression (11.34x) but fails to improve MAE (0.132). This stagnation may be driven by the "long-tail" effect: tokens added beyond $K = 2000$ appear too infrequently to receive sufficient gradient updates (of-

Table 10: Performance comparison with LLM-based models via textualized graph adaptation. These models exhibit significant performance degradation on pure structural tasks compared to our native tokenization.

| Model | ZINC (MAE ↓) | COIL-DEL (ACC ↑) |
|---|---|---|
| GraphGPT | 0.373 (0.021) | 5.6 (0.9) |
| LLAGA | 0.317 (0.016) | 12.5 (1.1) |
| **Our Method** | **0.131** (0.007) | **89.6** (0.2) |

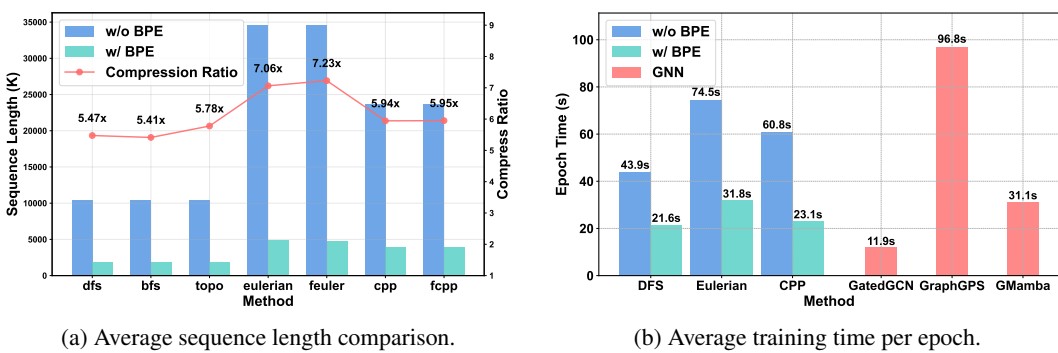

(a) Average sequence length comparison.      (b) Average training time per epoch.

Figure 6: Efficiency analysis on the `MolHIV` dataset.

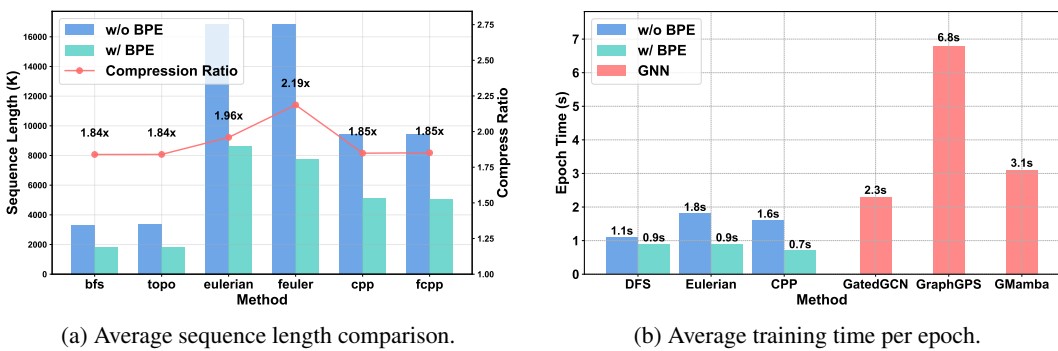

(a) Average sequence length comparison.      (b) Average training time per epoch.

Figure 7: Efficiency analysis on the `DD` dataset.

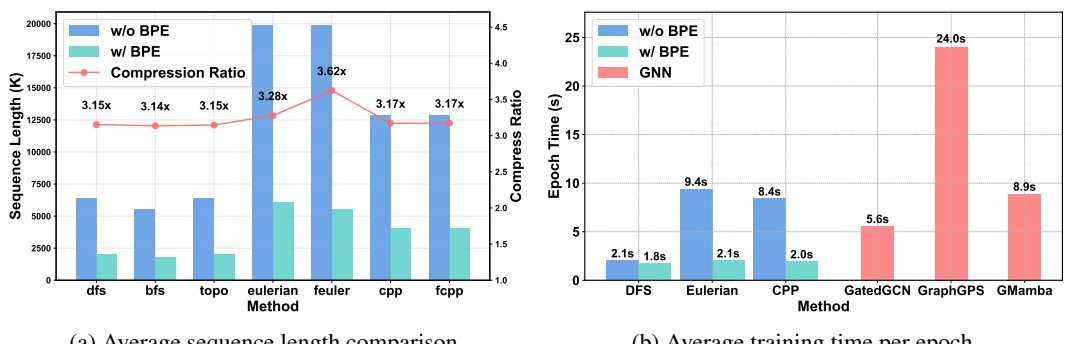

(a) Average sequence length comparison.      (b) Average training time per epoch.

Figure 8: Efficiency analysis on the `COLORS-3` dataset.

ten appearing fewer than 10 times across the entire dataset), thereby increasing model complexity without contributing to generalization.

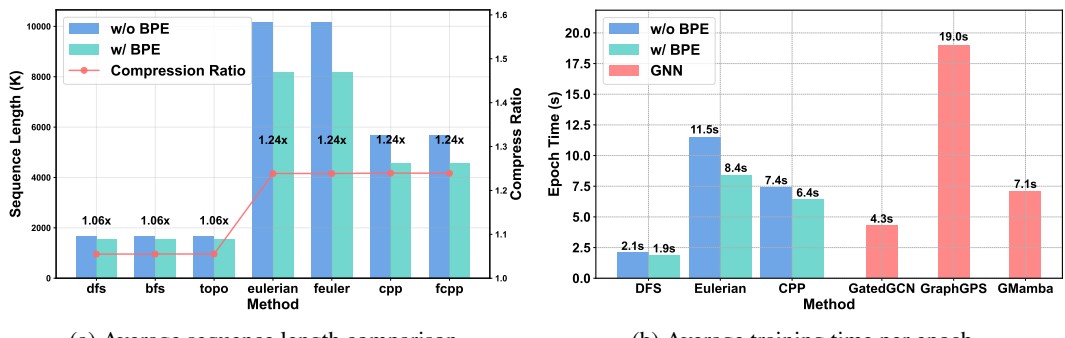

(a) Average sequence length comparison.

(b) Average training time per epoch.

Figure 9: Efficiency analysis on the `COIL-DEL` dataset.

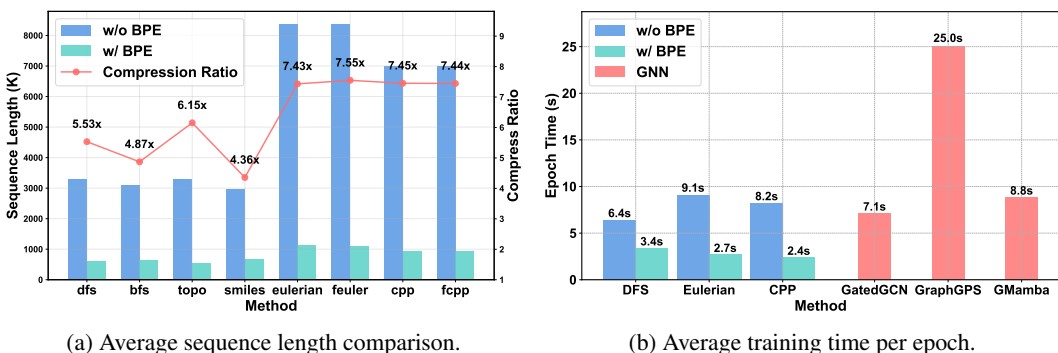

(a) Average sequence length comparison.

(b) Average training time per epoch.

Figure 10: Efficiency analysis on the `AQSOL` dataset.

**Choice of Frequency Guidance Unit.** The ablation confirms that Node-Edge-Node trigrams offer the superior trade-off, achieving the highest compression rate (10.84x) while maintaining linear complexity. The results in Table 15 show that simpler units like Node-Node bigrams perform suboptimally (10.71x) because they discard critical edge label information. Conversely, more complex units (2-hop/3-hop paths) surprisingly *reduce* compression (10.37x) despite their high computational cost ($O(E^3/N^2)$). This counter-intuitive result may be due to **data sparsity**. Specifically, as patterns grow longer, their exact recurrence becomes exponentially rarer, rendering the frequency map too sparse to provide robust global guidance.

**Serialization Strategy and Counting Tasks.** While most results are consistent with the main paper, the `COLORS-3` dataset presents a notable exception due to its unique task of counting nodes with specific colors. On this dataset, node-based serialization methods like DFS perform best as they visit each node exactly once, making the counting task trivial. In contrast, our default edge-traversal method (Eulerian) can visit a single node multiple times, creating ambiguity for direct counting.

To empirically validate this hypothesis, we conducted a controlled experiment on ZINC by simulating a node counting task (counting carbon atoms). As shown in Table 16, the edge-based serialization struggles with this task (21.6% accuracy), whereas switching to a node-based DFS traversal immediately resolves the failure (83.4% accuracy). This confirms that the limitation stems specifically from the mismatch between the edge-covering serialization strategy and the node-counting objective, rather than an inherent deficiency in the tokenizer or Transformer architecture. Despite this task-specific dynamic, a crucial finding remains: our **guided** serialization significantly outperforms unguided versions across general structural tasks, reinforcing the effectiveness of our guidance mechanism.

Table 11: Ablation with GT+GTE on additional datasets not covered in the main table. The best scores are shown in **bold**, the second-best are underlined, and standard deviations are in parentheses. A dash ("—") under SMILES indicates that the dataset either lacks SMILES representations or is not a molecular graph.

| Method | mutag acc↑ | | colors3 acc↑ | | dblp acc↑ | | aqsol mae↓ | | p-struct avg mae↓ | |
|---|---|---|---|---|---|---|---|---|---|---|
| | w | w/o | w | w/o | w | w/o | w | w/o | w | w/o |
| BFS | 79.2 (0.9) | 78.8 (0.6) | 73.4 (0.8) | 74.4 (0.3) | 91.8 (0.19) | 91.7 (0.22) | 0.701 (0.017) | 0.702 (0.008) | 0.2495 (0.0018) | 0.2494 (0.0005) |
| DFS | 81.9 (0.7) | 79.4 (1.5) | 99.9 (0.1) | **100.0** (0.0) | 92.8 (0.09) | 92.6 (0.12) | 0.684 (0.013) | 0.676 (0.011) | **0.2421** (0.0026) | 0.2468 (0.0040) |
| TOPO | 80.1 (0.8) | 78.4 (1.4) | 76.6 (1.1) | 98.3 (1.3) | 93.0 (0.09) | 92.5 (0.13) | 0.814 (0.004) | 0.737 (0.003) | 0.2626 (0.0022) | 0.2556 (0.0002) |
| Eulerian | 86.1 (0.9) | 84.3 (1.3) | 38.9 (1.0) | 44.7 (2.2) | 93.1 (0.09) | 91.7 (0.17) | | | 0.2514 (0.0027) | 0.2499 (0.0035) |
| Feuler | **90.1** (0.6) | 86.7 (0.9) | 41.0 (1.2) | 45.3 (0.8) | **93.6** (0.08) | 88.9 (0.14) | 0.621 (0.007) | 0.623 (0.017) | 0.2510 (0.0001) | 0.2548 (0.0004) |
| CPP | 87.7 (0.8) | 85.4 (0.9) | 36.9 (2.4) | 45.9 (1.2) | 90.9 (0.14) | 92.1 (0.21) | 0.654 (0.016) | 0.643 (0.003) | 0.2535 (0.0021) | 0.2482 (0.0005) |
| FCPP | 87.7 (0.4) | 86.0 (0.7) | 37.5 (1.4) | 45.6 (0.6) | 92.4 (0.08) | 92.3 (0.16) | 0.654 (0.017) | 0.633 (0.010) | 0.2537 (0.0039) | 0.2481 (0.0006) |
| SMILES | — | — | — | — | — | — | 0.741 (0.010) | **0.673** (0.010) | — | — |

Table 12: GT+BERT ablation of serialization orderings with and without BPE on main datasets. Best scores in **bold**, second-best underlined, std in parentheses. . A dash ("—") under SMILES indicates the dataset lacks SMILES or is not a molecular graph.

| Method | molhiv auc↑ | | coildel acc↑ | | p-func ap↑ | | zinc mae↓ | | qm9 mae↓ | |
|---|---|---|---|---|---|---|---|---|---|---|
| | w | w/o | w | w/o | w | w/o | w | w/o | w | w/o |
| BFS | 76.5 (0.6) | 76.1 (0.9) | 81.2 (0.9) | 80.1 (1.3) | 66.9 (0.9) | 63.2 (0.9) | 0.612 (0.009) | 0.961 (0.012) | 0.227 (0.011) | 0.324 (0.013) |
| DFS | 74.6 (0.4) | 79.5 (0.5) | 80.5 (0.4) | 79.8 (0.8) | **71.3** (0.3) | 67.5 (1.0) | 0.537 (0.008) | 0.976 (0.009) | 0.275 (0.009) | 0.281 (0.014) |
| TOPO | 72.2 (0.6) | 77.9 (0.8) | **82.6** (0.8) | 81.4 (1.2) | 65.3 (0.5) | 54.6 (1.0) | 0.711 (0.011) | 1.034 (0.012) | 0.277 (0.010) | 0.266 (0.011) |
| Eulerian | 83.7 (0.7) | 83.9 (1.0) | 72.1 (0.6) | 69.9 (2.9) | 67.3 (0.9) | 64.1 (1.6) | 0.304 (0.008) | 0.396 (0.012) | **0.104** (0.006) | 0.127 (0.007) |
| Feuler | 82.6 (0.4) | 81.8 (0.5) | 74.1 (0.3) | 76.1 (0.9) | 68.5 (1.0) | 65.4 (1.1) | **0.241** (0.006) | 0.432 (0.011) | 0.122 (0.004) | 0.128 (0.005) |
| CPP | **85.0** (0.3) | 82.8 (0.5) | 72.5 (0.5) | 83.3 (0.7) | 65.6 (0.6) | 59.5 (1.5) | 0.319 (0.005) | 0.464 (0.008) | 0.115 (0.004) | 0.131 (0.006) |
| FCPP | 83.2 (0.3) | 82.2 (0.6) | 68.9 (0.3) | 78.3 (1.0) | 65.5 (1.8) | 61.0 (1.2) | 0.316 (0.005) | 0.467 (0.007) | 0.107 (0.005) | 0.132 (0.007) |
| SMILES | — | — | — | — | — | — | 0.273 (0.014) | 0.320 (0.007) | 0.117 (0.014) | 0.120 (0.016) |

## D.4 SCALABILITY ANALYSIS ON LARGE-SCALE OGB DATASETS

To understand the computational characteristics of our pipeline on large graphs, we profile each stage on OGB datasets containing millions of edges (ogbn-arxiv, ogbg-code2, and ogbn-products). All measurements use the Frequency-Guided Eulerian circuit for serialization and are performed on a single CPU thread, with results normalized to *Time per 1 Million Nodes*.

Table 17 reveals a clear separation between the two stages. BPE encoding introduces negligible overhead, completing in tens of milliseconds per million nodes and remaining stable across datasets regardless of graph density. The computational cost of the pipeline is thus dominated by the serialization stage, whose runtime scales with edge density and serialization method. This decomposition highlights a practical advantage of our modular design: since the tokenization stage (BPE) is lightweight, the overall efficiency can be improved by selecting a serialization method suited to the scale and structure of the target graphs.

Table 13: GT+BERT ablation on additional datasets (Appendix). Best scores in **bold**, second-best underlined, std in parentheses. A dash ("—") under SMILES indicates the dataset lacks SMILES or is not a molecular graph.

| Method | mutag acc↑ | | colors3 acc↑ | | dblp acc↑ | | aqsol mae↓ | | p-struct avg mae↓ | |
|---|---|---|---|---|---|---|---|---|---|---|
| | w | w/o | w | w/o | w | w/o | w | w/o | w | w/o |
| BFS | 76.1 (0.6) | 76.4 (0.7) | 72.7 (0.9) | 67.3 (4.1) | 91.9 (0.17) | 91.3 (0.26) | 0.851 (0.002) | 0.844 (0.002) | 0.2477 (0.0014) | 0.2620 (0.0021) |
| DFS | 79.6 (0.5) | 77.3 (0.6) | **98.3** (0.1) | 87.5 (4.4) | 92.6 (0.08) | 92.4 (0.13) | 0.831 (0.002) | 0.852 (0.002) | 0.2526 (0.0036) | 0.2550 (0.0012) |
| TOPO | 74.9 (0.5) | 71.2 (0.7) | 60.4 (2.3) | 66.6 (2.2) | 92.7 (0.08) | 92.7 (0.11) | 0.810 (0.002) | 0.840 (0.002) | 0.2578 (0.0011) | 0.2648 (0.0026) |
| Eulerian | 85.5 (0.8) | 82.0 (1.1) | 37.8 (1.4) | 40.7 (0.6) | 93.1 (0.04) | 91.8 (0.14) | **0.648** (0.002) | 0.677 (0.003) | 0.2522 (0.0012) | 0.2700 (0.0049) |
| Feuler | **87.5** (0.4) | 84.1 (0.8) | 38.5 (1.1) | 40.3 (1.0) | **93.2** (0.06) | 88.5 (0.11) | **0.648** (0.002) | 0.685 (0.004) | **0.2476** (0.0010) | 0.2615 (0.0045) |
| CPP | 85.9 (0.6) | 83.6 (0.9) | 36.5 (2.0) | 43.5 (1.2) | 90.5 (0.11) | 91.7 (0.22) | 0.651 (0.018) | 0.690 (0.018) | 0.2547 (0.0018) | 0.2734 (0.0022) |
| FCPP | 85.7 (0.5) | 85.0 (0.9) | 37.1 (0.6) | 44.5 (1.1) | 91.8 (0.13) | 91.9 (0.21) | 0.670 (0.005) | 0.695 (0.003) | 0.2541 (0.0029) | 0.2681 (0.0051) |
| SMILES | — | — | — | — | — | — | 0.746 (0.003) | 0.783 (0.003) | — | — |

Table 14: Impact of BPE merge count ($K$) on compression ratio and model performance (ZINC). $K = 2000$ achieves the best balance; larger vocabularies yield diminishing compression returns and do not improve performance due to token rarity.

| Merge Count ($K$) | Compression Ratio | Performance (MAE ↓) |
|---|---|---|
| 0 (Raw) | 1.00x | 0.171 (0.013) |
| 100 | 2.86x | 0.144 (0.012) |
| 500 | 5.37x | 0.137 (0.010) |
| 1000 | 8.11x | 0.131 (0.011) |
| **2000** | **10.84x** | **0.131** (0.007) |
| 5000 | 11.34x | 0.132 (0.009) |

Table 15: Impact of different statistical units on BPE compression ratio (ZINC). 'Collect Cost' denotes the complexity of gathering statistics. Trigrams achieve the highest compression with linear overhead.

| Unit | Collect Cost | Compression Ratio |
|---|---|---|
| W/O Guidance | 0 | 10.46x |
| Node-Node Bigram | $O(E)$ | 10.71x |
| Node-Edge Bigram | $O(E)$ | 10.72x |
| **Node-Edge-Node Trigram** | **O(E)** | **10.84x** |
| Multi-hop Path (2 hop) | $O(E^2/N)$ | 10.56x |
| Multi-hop Path (3 hop) | $O(E^3/N^2)$ | 10.37x |

Table 16: Performance on simulated node counting tasks (ZINC). Switching from edge-based to node-based serialization resolves the counting failure, confirming our analysis regarding the COLORS-3 dataset.

| Serialization Method | Accuracy (w/ BPE) | Accuracy (w/o BPE) |
|---|---|---|
| Edge-based (Feuler) | 21.6% | 24.8% |
| **Node-based (DFS)** | **83.4%** | **88.4%** |

Table 17: Efficiency of different components of our method on large-scale OGB datasets (Normalized to **Time per 1 Million Nodes**).

| Dataset | Serialization (ms) | BPE Encoding (ms) |
|---|---|---|
| ogbn-arxiv | 14,948 | 57 |
| ogbg-code2 | 4,108 | 74 |
| ogbn-products | 29,670 | 56 |

