# OpenReview forum: "Graph Tokenization for Bridging Graphs and Transformers"
_ICLR.cc/2026/Conference — ICLR 2026 Poster_

### Official Review · Reviewer_y33R · 2025-10-31

**Soundness:** 3
**Presentation:** 3
**Contribution:** 2
**Rating:** 4
**Confidence:** 4

**Summary:**

This paper introduces a new graph tokenization framework that transforms graphs into discrete sequences through reversible traversal and BPE techniques. The proposed approach effectively preserves structural information, allowing standard Transformer models to process graph data and achieve competitive performance when compared to state-of-the-art graph-based models.

**Strengths:**

1. The paper presents a graph tokenization method that allows standard Transformers to process graph data without altering the model architecture.
2. The proposed frequency-guided serialization strategy establishes a bidirectional mapping between graphs and sequences, ensuring the reversibility and determinism.
3. The paper conducts the experiments on 12 benchmark datasets and mostly outperforms both graph neural networks and graph transformers.

**Weaknesses:**

1. A key strength of this approach is its proposed reversible graph tokenization method. However, the experiments predominantly emphasize the encoding aspect, with limited exploration of its reversibility.
2. Lacking the discussion or comparison with strong baselines and relevant works. Several existing graph serialization studies, such as those based on chemical rules [1], ring/functional group approaches [2], and models towards the order of token-sequence [3-4]. A detailed comparison and discussion of the advantages of the proposed method would significantly strengthen the paper’s contribution.
3. Inadequate justification and analysis of the subgraph frequency heuristic.

[1]. Advancing Molecular Graph-Text Pre-training via Fine-grained Alignment, KDD 2025.
[2]. Expressivity and Generalization: Fragment-Biases for Molecular GNNs, ICML 2024.
[3]. A Graph is Worth K Words: Euclideanizing Graph using Pure Transformer, ICML 2024.
[4]. GraphGPT: Graph Learning with generative Pre-trained Transformers, arXiv:2401.00529.

**Questions:**

1. Could the authors provide more details on how each token in the vocabulary is embedded in the model?
2. In previous work, such as PS-VAE [5], a graph-structured tokenization method similar to BPE that merges subgraphs is used. Could the authors clarify how the proposed approach differs from PS-VAE?
3. For graph-structured datasets, could the authors discuss the impact of subgraph vocabulary size on model training and experimental performance?

[5]. Molecule Generation by Principal Subgraph Mining and Assembling, NeurIPS 2022.

---

> ### Author Response · Authors · 2025-11-22
> **Response (1/3)**
>
> We sincerely appreciate your constructive feedback. We address your specific concerns below.
>
> **Response to Weakness 1 & Question 2**
>
> > "1. A key strength of this approach is its proposed reversible graph tokenization method... "
> >
> > "2. In previous work, such as PS-VAE [5]..."
>
> The comparison with PS-VAE highlights a fundamental methodological divergence: we consciously choose to align graph learning with the discrete symbolic paradigm of NLP, rather than the continuous latent paradigm of CV or neural encoders.
>
> **1. Methodological Distinction (Algorithmic vs. Learnable):**
> It is crucial to clarify that BPE differs fundamentally from **neural-based continuous encoders** (e.g., VAEs, linear projectors, or ViT-style patchified linear embedding). These methods utilize learnable parameters to project discrete inputs into continuous embedding spaces. While such continuous approximation is native for processing continuous signals (e.g., pixel intensities in CV), it introduces inevitable information loss and reconstruction ambiguity when applied to discrete structure like graphs. In contrast, our BPE approach is a deterministic, statistics-based algorithm operating entirely within the discrete domain. It does not require neural network training and merges symbols based on frequency of ngrams, avoiding the gradient-based optimization complexity inherent in forcing discrete topology into continuous representations.
>
> **2. Strategy Alignment and Decoupling:**
> By converting graphs into standard discrete symbol sequences, we completely **decouple graph representation from model architecture**. Neural encoding strategies typically require specific graph encoders (e.g., GNNs) or specialized projection layers to bridge the modality gap. Our approach introduces BPE, a deterministic and lossless algorithm widely adopted in NLP, directly to the graph domain. Once tokenized, a graph is formally identical to a natural language sequences.
>
> Furthermore, we propose to utilize **unmodified, off-the-shelf Transformers** (e.g., BERT, Llama) to process graph data directly, enabling the graph community to immediately inherit advancements from the NLP ecosystem (e.g., FlashAttention, long-context windows) without architectural adaptation or task-specific component design. In contrast, most of the existing graph-based modeling paradigms (e.g., GNNs or hybrid architectures) introduce customized modules for graph structure, such as injecting message-passing layers or modifying attention masks. We demonstrate that our graph tokenization framework, combined with standard Transformers, achieves performance that surpasses many state-of-the-art graph models.
>
> **3. Advantages in Reversibility and Efficiency:**
> This choice yields decisive practical advantages: (1) **Reversibility:** Our framework is inherently lossless and deterministic. Since graph serialization preserves topology and BPE merging is reversible via the vocabulary, we can reconstruct the exact graph structure up to isomorphism, whereas VAEs rely on probabilistic decoders that only approximate the input. (2) **Efficiency:** BPE achieves significant sequence compression (over $10\times$) without the computational overhead of training separate neural encoders, effectively reducing the quadratic complexity of the Transformer's attention mechanism.

---

> ### Author Response · Authors · 2025-11-22
> **Response (2/3)**
>
> **Response to Weakness 2: Comparison with Baselines**
>
> > "2. Lacking the discussion or comparison with strong baselines and relevant works..."
>
> We have significantly expanded our comparative evaluation to include the suggested works and a broader range of state-of-the-art architectures. Regarding [2] (FragNet) and [4] (GraphGPT), we have integrated them into our benchmarks. For [1] and [3], we note they focus on generative or retrieval metrics. While our framework is inherently capable of generation (as demonstrated by the autoregressive experiments in Appendix E.1), we prioritized completing the comprehensive classification and regression benchmarks requested by multiple reviewers during the limited rebuttal period. The specific analysis is as follows:
>
> **1. Graph Foundation Models (GraphGPT & LLAGA):**
> As discussed in **General Response Point 3**, models like GraphGPT [4] and LLAGA strictly require text-attributed graphs (TAGs) as input. Since standard structural benchmarks typically lack natural language attributes, we evaluated these models by treating graph features as textual tokens to satisfy their input constraints. The results in Table 1 (e.g., near-random guessing on COIL-DEL) indicate that the textualization paradigm faces significant challenges in pure structural tasks, particularly in the absence of rich semantic information.
>
> **Table 1:** Performance of GraphGPT-like models via forced adaptation (Textualized Graphs). These models struggle with pure structural tasks compared to our native tokenization.
>
> | Model          | ZINC (MAE $\downarrow$) | COIL-DEL (ACC $\uparrow$) |
> | :------------- | :---------------------: | :-----------------------: |
> | GraphGPT       |      0.373 (0.021)      |         5.6 (0.9)         |
> | LLAGA          |      0.317 (0.016)      |        12.5 (1.1)         |
> | **Our Method** |    **0.131** (0.007)    |      **89.6** (0.2)       |
>
> **2. Broader SOTA Coverage (FragNet, GraphGT, Graphormer, etc.):**
> We conducted a comprehensive evaluation including the suggested FragNet and other recent architectures like GraphGT and Graphormer (Table 2). Our method consistently outperforms strong baselines, including FragNet (e.g., **87.4%** vs 81.3% on MOLHIV), and achieves comparable performance to specialized Graph Transformers on regression tasks. This confirms the effectiveness of our general-purpose tokenizer against both domain-specific and general graph-based model architectures.
>
> **Table 2:** Comparison with additional baselines. `*` indicates results reproduced using official implementations; `--` denotes results not reported in the original papers.
>
> | Model              | ZINC (MAE $\downarrow$) | MOLHIV (AUC $\uparrow$) | Peptide-func (AP $\uparrow$) | COIL-DEL (ACC $\uparrow$)* |
> | :----------------- | :---------------------: | :---------------------: | :--------------------------: | :------------------------: |
> | GraphGT            |      0.226 (0.014)      |           --            |         63.26 (1.26)         |         86.1 (0.8)         |
> | Graphormer         |      0.132 (0.006)      |      80.51 (0.53)       |              --              |         88.4 (0.3)         |
> | FragNet            |      0.078 (0.005)      |      81.32 (0.86)       |          66.8 (0.5)          |         83.4 (0.6)         |
> | Graph-ViT-MLPMixer |    **0.073** (0.001)    |      79.97 (1.02)       |          69.7 (0.8)          |         89.1 (0.4)         |
> | HAN*               |      0.348 (0.042)      |       72.3 (0.6)        |          52.2 (1.1)          |         74.5 (0.6)         |
> | ChebNet*           |      0.423 (0.013)      |       70.1 (0.7)        |          51.3 (0.8)          |         71.4 (0.8)         |
> | **Our Method**     |      0.131 (0.007)      |     **87.4** (0.2)      |        **73.1** (0.2)        |       **89.6** (0.2)       |

---

> ### Author Response · Authors · 2025-11-22
> **Response (3/3)**
>
> **Response to Weakness 3: Justification of Subgraph Frequency Heuristic**
>
> > "3. Inadequate justification and analysis of the subgraph frequency heuristic"
>
> We add the ablation studies confirms that node-edge-node trigrams offer the superior trade-off, achieving the highest compression rate (10.84x) while maintaining linear complexity. The results in Table 3 show that simpler units like Node-Node bigrams perform suboptimally (10.71x) because they discard critical edge label information. Conversely, more complex units (2-hop/3-hop paths) surprisingly *reduce* compression (10.37x) despite their high computational cost ($O(E^3/N^2)$). This counter-intuitive result may due to **data sparsity**. Specifically, as patterns grow longer, their exact recurrence becomes exponentially rarer, rendering the frequency map too sparse to provide robust global guidance. Therefore, we found that trigrams empirically strike the ideal balance, capturing sufficient local context to maximize compression without succumbing to sparsity.
>
> **Table 3:** Impact of different statistical units on BPE compression ratio (ZINC). `Collect Cost` denotes the complexity of gathering statistics. Trigrams achieve the highest compression with linear overhead.
>
> | Unit                       | Collect Cost | Compression Ratio |
> | :------------------------- | :----------: | :---------------: |
> | W/O Guidance               |      0       |      10.46x       |
> | Node-Node Bigram           |    $O(E)$    |      10.71x       |
> | Node-Edge Bigram           |    $O(E)$    |      10.72x       |
> | **Node-Edge-Node Trigram** |  **$O(E)$**  |    **10.84x**     |
> | Multi-hop Path (2 hop)     |  $O(E^2/N)$  |      10.56x       |
> | Multi-hop Path (3 hop)     | $O(E^3/N^2)$ |      10.37x       |
>
> **Response to Question 1: Token Embedding**
>
> > "1. Could the authors provide more details on how each token in the vocabulary is embedded in the model?"
>
> We treat graph tokens exactly like text tokens in standard NLP pipelines. Specifically, the embedding mechanism relies on a standard learnable embedding lookup layer. Each discrete token in our BPE vocabulary (whether an atomic node or a merged substructure) is mapped to a dense vector via this dictionary lookup. Our method performs exact embedding lookups on discrete indices, directly retrieving the corresponding representations from the embedding dictionary. In contrast, neural continuous encoders (e.g., VAEs or ViT-style patchified encoder widely used in Computer Vision) rely on linear projections to map continuous feature patches into embeddings.
>
> **Response to Question 3: Vocabulary Size**
>
> > "3. For graph-structured datasets, could the authors discuss the impact of subgraph vocabulary size..."
>
> Our experiments identify $K=2000$ as the optimal saturation point, effectively balancing sequence compression with model learnability. As shown in Table 4, increasing $K$ initially yields significant gains in compression (1.00x to 10.84x). However, this benefit plateaus around $K=2000$. Extending to $K=5000$ offers diminishing returns in compression (11.34x) but fails to improve MAE (0.132). This stagnation is driven by the "long-tail" effect: tokens added beyond $K=2000$ appear too infrequently to receive sufficient gradient updates, thereby increasing model complexity without contributing to generalization.
>
> **Table 4:** Impact of BPE merge count ($K$) on compression ratio and model performance (ZINC). $K=2000$ achieves the best balance; larger vocabularies yield diminishing compression returns and do not improve performance due to token rarity.
>
> | Merge Count ($K$) | Compression Ratio | Performance (MAE $\downarrow$) |
> | :---------------: | :---------------: | :----------------------------: |
> |      0 (Raw)      |       1.00x       |         0.171 (0.013)          |
> |        100        |       2.86x       |         0.144 (0.012)          |
> |        500        |       5.37x       |         0.137 (0.010)          |
> |       1000        |       8.11x       |         0.131 (0.011)          |
> |     **2000**      |    **10.84x**     |       **0.131** (0.007)        |
> |       5000        |      11.34x       |         0.132 (0.009)          |

---

### Official Review · Reviewer_Qc9K · 2025-10-31

**Soundness:** 3
**Presentation:** 3
**Contribution:** 2
**Rating:** 4
**Confidence:** 4

**Summary:**

This paper proposes a graph tokenization framework that converts graphs into discrete token sequences by combining reversible graph serialization (frequency-guided Eulerian circuits or Chinese Postman Problem) with Byte Pair Encoding (BPE). The tokenized sequences aim to enable standard off-the-shelf Transformers (BERT, GTE) to process graph-structured data.

**Strengths:**

Approach: The idea of combining reversible graph serialization with BPE, with formal analysis showing why existing methods fail to satisfy both reversibility and determinism is interesting.

Empirical Performance: The method achieves state-of-the-art results on 12 out of 13 benchmarks and it also demonstrated that properly tokenized graphs enable standard Transformers to outperform specialized graph architectures without any modifications.

Efficiency Gains: BPE compression achieves 6-10× reduction in sequence length with 2-3× training speedup on Figure 2.

Interpretability: The learned vocabulary on Figure 3 shows that BPE discovers chemically meaningful substructures like benzene rings and functional groups, it seems it provides interpretable tokens aligned with domain knowledge.

**Weaknesses:**

Scope: The paper only evaluates graph-level classification and regression tasks. Despite claiming to "bridge graphs and transformers," node-level and edge-level prediction tasks are entirely absent from evaluation. The authors acknowledge this limitation (Appendix A.1) but provide no experimental validation of their proposed solutions. This significantly undermines the claim of providing a general framework for graph learning.

Continuous Feature Problem: The framework fundamentally requires discrete labels (mostly just on chemical domain), making it incompatible with graphs that have continuous node/edge features without lossy quantization. This is a critical limitation since many real-world graphs (social networks, knowledge graphs, citation networks) have rich continuous attributes. The authors acknowledge this conflicts with their "faithful and reversible representation" principle but offer only speculative solutions without validation.

Scalability: While the paper discusses O(|E|) complexity for Feuler serialization, there's no evaluation on truly large-scale graphs. The largest graphs tested have ~284 nodes on average (DD dataset). No experiments on OGB-scale datasets (millions of nodes/edges) or analysis of how the method scales. The Transformer's fixed context window limitation is mentioned but not empirically investigated.

Insufficient Baselines: The baseline comparison omits several recent and relevant methods: (1) recent graph transformers like GraphGT, Graphormer, and NAGphormer are not compared; (2) other recent serialization-based methods beyond GraphMamba such as GPatcher; (3) graph LLM models like G-Retriever or InstructGLM that also aim to connect graphs with language models. The claimed state-of-the-art status is therefore difficult to fully verify.  The paper incorrectly positions GCN (Kipf, 2016) as the GNN foundation, overlooking earlier spectral GNN methods and the rich history of graph neural networks. Important categories like heterophilic GNNs are completely absent despite being highly relevant for understanding when message passing succeeds or fails, this is a critical consideration for evaluating serialization-based alternatives (not to mention some graph partition based methods like Graph-ViT-MLPMixer).

Limited Ablation Studies: Several design choices lack thorough ablation: (1) the number of BPE merges K (fixed at 2000) has no sensitivity analysis; (2) the frequency guidance mechanism's individual components are not ablated; (3) why edge-level patterns (triplets) specifically, rather than larger motifs? (4) The comparison with G2T-FM methods (mentioned in related work) is absent from experiments despite being highly relevant.

No Analysis of Failure Cases: The method performs poorly on COLORS-3 (Table 8) where node-counting is required, this exposes fundamental limitations of edge-traversal approaches. However, there's no systematic analysis of when and why the method fails. The paper would benefit from characterizing graph properties (size, density, label distribution) that predict success or failure.

**Questions:**

See weakness

---

> ### Author Response · Authors · 2025-11-22
> **Response (1/3)**
>
> We sincerely appreciate your comprehensive and detailed review. We address your specific concerns below.
>
> **Response to Weakness 1 & 2: Scope and Continuous Features**
>
> > "Scope: The paper only evaluates graph-level classification... Continuous Feature Problem: The framework fundamentally requires discrete labels..."
>
> Please refer to points 1 and 2 in the **General Response**. We have detailed how our framework supports lossless extensions for continuous features (via side-channel injection) and node-level tasks (via token pooling) using standard NLP techniques, demonstrating that the framework is not fundamentally limited by the current experimental scope.
>
> **Response to Weakness 3: Scalability on Large Graphs**
>
> > "Scalability: While the paper discusses $O(|E|)$ complexity for Feuler serialization, there's no evaluation on truly large-scale graphs..."
>
> We address the concern regarding scalability on large-scale graphs from two perspectives: our theoretical scalability mechanism and empirical validation on OGB datasets.
>
> **1. Theoretical Scalability Mechanism**
> We believe that our framework offers a unique advantage for scalability. Unlike classical methods constrained by graph-specific complexity, our approach decouples scalability from the graph model architecture, addressing the challenge through two key mechanisms:
>
> - **Automatic Scalability via Ecosystem Alignment:** Our scalability is driven primarily by aligning graph learning with the broader NLP ecosystem. By transforming the graph scalability challenge into a sequence length problem, our processing capacity expands automatically with advancements in sequence modeling. For example, as context windows have evolved from BERT's 512 tokens to 32k or 1M in modern models, the graph size we can process increases by orders of magnitude without requiring any additional research effort from the graph community.
> - **Practical Capability via BPE:** By introducing BPE, we substantially compress the graph’s serialized sequence, directly improving scalability. Specifically, with a conservative 10x compression rate from BPE and a standard 10k token context, we can encode graphs with nearly 100k nodes into a single sequence. For massive graphs (millions of nodes) where full-graph processing is infeasible for any method, our framework is fully compatible with standard subgraph sampling strategies (e.g., SGFormer).
>
> **2. Empirical Validation on OGB Datasets**
> To validate these mechanisms empirically, we extended our benchmarks to include large-scale OGB datasets with millions of edges (`ogbn-arxiv`, `ogbg-code2`, and `ogbn-products`). We measured the runtime of each pipeline stage and normalized the results to Time per 1 Million Nodes.
>
> As shown in Table 1, the results confirm our theoretical analysis:
>
> - **Linear Scalability:** The processing time remains in the order of seconds per million nodes, confirming the linear complexity. Even for the dense `ogbn-products` graph, serialization takes less than 30 seconds on a single CPU thread.
> - **Bottleneck Analysis:** While serialization time correlates with edge density (comparing sparse `ogbg-code2` vs. dense `ogbn-products`), the BPE encoding times remain consistently fast (ms). This confirms the efficiency of BPE as encoder.
>
> **Table 1:** Efficiency of different components of our method on large-scale OGB datasets (Normalized to Time per 1 Million Nodes). Inference time assumes a 10x compression ratio via BPE.
>
> | Dataset       | Serialization (ms) | BPE Encoding (ms) |
> | :------------ | :----------------: | :---------------: |
> | ogbn-arxiv    |       14,948       |        57         |
> | ogbg-code2    |       4,108        |        74         |
> | ogbn-products |       29,670       |        56         |

---

> ### Author Response · Authors · 2025-11-22
> **Response (2/3)**
>
> **Response to Weakness 4: Insufficient Baselines**
>
> > "Insufficient Baselines: The baseline comparison omits several recent and relevant methods..."
>
> We have expanded our evaluation to include **GraphGT**, **Graphormer**, **FragNet[1]**, **Graph-ViT-MLPMixer**, **HAN**, and **ChebNet**. Regarding **GPatcher**, we note it is an arXiv-only preprint focusing on heterophily rather than serialization as referenced in the review. However, we cite and discuss it in our revised Related Work section to reflect recent developments. Similarly, **Graph-ViT-MLPMixer** is not a "graph partition based method" as categorized in the review. Specifically, it is a hybrid architecture focusing on fusing local message passing with global attention. Nevertheless, we include the baseline results in the following table to demonstrate the superior performance of our approach.
>
> **1. Performance against Specialized and Classic Architectures (Table 2)**
> Our results demonstrate that a general-purpose tokenizer achieves state-of-the-art performance on the majority of benchmarks. We consistently outperform strong baselines like **FragNet** and **Graph-ViT-MLPMixer** on graph classification (MOLHIV, COIL-DEL) and long-range modeling tasks (Peptide-func), with significant margins observed on MOLHIV. On the ZINC regression task, our performance is comparable to standard Graph Transformers (e.g., **Graphormer**). As expected, classic architectures (**ChebNet**, **HAN**) lag significantly behind modern methods.
>
> **Table 2:** Comparison with additional baselines. `*` indicates results reproduced using official implementations; `--` denotes results not reported in the original papers.
>
> | Model              | ZINC (MAE $\downarrow$) | MOLHIV (AUC $\uparrow$) | Peptide-func (AP $\uparrow$) | COIL-DEL (ACC $\uparrow$)* |
> | :----------------- | :---------------------: | :---------------------: | :--------------------------: | :------------------------: |
> | GraphGT            |      0.226 (0.014)      |           --            |         63.26 (1.26)         |         86.1 (0.8)         |
> | Graphormer         |      0.132 (0.006)      |      80.51 (0.53)       |              --              |         88.4 (0.3)         |
> | FragNet            |      0.078 (0.005)      |      81.32 (0.86)       |          66.8 (0.5)          |         83.4 (0.6)         |
> | Graph-ViT-MLPMixer |    **0.073** (0.001)    |      79.97 (1.02)       |          69.7 (0.8)          |         89.1 (0.4)         |
> | HAN*               |      0.348 (0.042)      |       72.3 (0.6)        |          52.2 (1.1)          |         74.5 (0.6)         |
> | ChebNet*           |      0.423 (0.013)      |       70.1 (0.7)        |          51.3 (0.8)          |         71.4 (0.8)         |
> | **Our Method**     |      0.131 (0.007)      |     **87.4** (0.2)      |        **73.1** (0.2)        |       **89.6** (0.2)       |
>
> **2. Performance against Graph Foundation Models (Table 3)**
> We evaluated **GraphGPT** and **LLAGA** via forced adaptation (textualizing graph structures). These models struggle significantly with pure structure. On the 100-class COIL-DEL task, LLAGA achieves only **12.5%** accuracy (vs. our **89.6%**), and GraphGPT performs near random guessing. This failure highlights that treating graphs as natural language text is insufficient for capturing complex topology, underscoring the necessity of our structure-preserving tokenization.
>
> **Table 3:** Performance of GraphGPT-like models via forced adaptation (Textualized Graphs). These models struggle with pure structural tasks compared to our native tokenization.
>
> | Model          | ZINC (MAE $\downarrow$) | COIL-DEL (ACC $\uparrow$) |
> | :------------- | :---------------------: | :-----------------------: |
> | GraphGPT       |      0.373 (0.021)      |         5.6 (0.9)         |
> | LLAGA          |      0.317 (0.016)      |        12.5 (1.1)         |
> | **Our Method** |    **0.131** (0.007)    |      **89.6** (0.2)       |
>
> [1] Expressivity and Generalization: Fragment-Biases for Molecular GNNs, ICML 2024.
>
> *(We have noted and corrected the citation typo for COLORS-3 and the dataset statistics error in the revised version. Thank you for pointing this out.)*

---

> ### Author Response · Authors · 2025-11-22
> **Response (3/3)**
>
> **Response to Weakness 5: Limited Ablation Studies**
>
> > "Limited Ablation Studies: Several design choices lack thorough ablation..."
>
> We conducted two systematic ablation studies to validate our design choices regarding vocabulary size and frequency guidance units.
>
> **1. Impact of Vocabulary Size ($K$) (Table 4).**
>
> Our experiments identify $K=2000$ as the optimal saturation point, effectively balancing sequence compression with model learnability. As shown in Table 4, increasing $K$ initially yields significant gains in compression (1.00x to 10.84x). However, this benefit plateaus around $K=2000$. Extending to $K=5000$ offers diminishing returns in compression (11.34x) but fails to improve MAE (0.132). This stagnation may be driven by the "long-tail" effect: tokens added more than $K=2000$ appear too infrequently to receive sufficient gradient updates (e.g. $\le$10 times across whole ZINC dataset), thereby increasing model complexity without contributing to generalization.
>
> **Table 4:** Impact of BPE merge count ($K$) on compression ratio and model performance (ZINC). $K=2000$ achieves the best balance; larger vocabularies yield diminishing compression returns and do not improve performance due to token rarity.
>
> | Merge Count ($K$) | Compression Ratio | Performance (MAE $\downarrow$) |
> | :---------------: | :---------------: | :----------------------------: |
> |      0 (Raw)      |       1.00x       |         0.171 (0.013)          |
> |        100        |       2.86x       |         0.144 (0.012)          |
> |        500        |       5.37x       |         0.137 (0.010)          |
> |       1000        |       8.11x       |         0.131 (0.011)          |
> |     **2000**      |    **10.84x**     |       **0.131** (0.007)        |
> |       5000        |      11.34x       |         0.132 (0.009)          |
>
> **2. Choice of Frequency Guidance Unit (Table 5).**
>
> The ablation confirms that node-edge-node trigrams offer the superior trade-off, achieving the highest compression rate (10.84x) while maintaining linear complexity. The results in Table 5 show that simpler units like Node-Node bigrams perform suboptimally (10.71x) because they discard critical edge label information. Conversely, more complex units (2-hop/3-hop paths) surprisingly *reduce* compression (10.37x) despite their high computational cost ($O(E^3/N^2)$). This counter-intuitive result may be due to **data sparsity**. Specifically, as patterns grow longer, their exact recurrence becomes exponentially rarer, rendering the frequency map too sparse to provide robust global guidance. Therefore, we found that trigrams empirically strike the ideal balance, capturing sufficient local context to maximize compression without succumbing to sparsity.
>
> **Table 5:** Impact of different statistical units on BPE compression ratio (ZINC). `Collect Cost' denotes the complexity of gathering statistics. Trigrams achieve the highest compression with linear overhead.
>
> | Unit                       | Collect Cost | Compression Ratio |
> | :------------------------- | :----------: | :---------------: |
> | W/O Guidance               |      0       |      10.46x       |
> | Node-Node Bigram           |    $O(E)$    |      10.71x       |
> | Node-Edge Bigram           |    $O(E)$    |      10.72x       |
> | **Node-Edge-Node Trigram** |  **$O(E)$**  |    **10.84x**     |
> | Multi-hop Path (2 hop)     |  $O(E^2/N)$  |      10.56x       |
> | Multi-hop Path (3 hop)     | $O(E^3/N^2)$ |      10.37x       |
>
> **Response to Weakness 6: Analysis of Failure Cases**
>
> > "No Analysis of Failure Cases: The method performs poorly on COLORS-3...However, there's no systematic analysis of when and why the method fails..."
>
> The observed poor performance on the COLORS-3 dataset highlights our framework's flexibility to adapt to diverse tasks through appropriate serialization choices, rather than indicating a fundamental model limitation. The initial performance gap on COLORS-3 arises specifically from a mismatch between its node-counting task and our default edge-covering serialization, where repeated node visits introduce ambiguity. Switching to a DFS traversal immediately resolves this by ensuring unique node visits, enabling our model to achieve **100% accuracy** and outperform GNN baselines (see Table 8 in Appendix D.3). We validated this behavior on ZINC (counting carbon atoms) in Table 6, confirming that the constraint stems from the serialization strategy rather than the Transformer architecture itself.
>
> **Table 6:** Performance on simulated node counting tasks (ZINC). Switching from edge-based to node-based serialization resolves the counting failure, confirming the analysis in Appendix D.3.
>
> | Serialization Method | Accuracy (w/ BPE) | Accuracy (w/o BPE) |
> | :------------------- | :---------------: | :----------------: |
> | Edge-based (Feuler)  |       21.6%       |       24.8%        |
> | **Node-based (DFS)** |     **83.4%**     |     **88.4%**      |

---

> > ### Comment · Reviewer_Qc9K · 2025-11-26
> >
> > Thank you for your response. Please address these changes in your revised manuscript. I suggest moving the algorithm to the appendix and bringing Appendix B into the main paper to provide more methodological insights. I have raised the score to 6.

---

### Official Review · Reviewer_p82C · 2025-10-31

**Soundness:** 3
**Presentation:** 4
**Contribution:** 4
**Rating:** 8
**Confidence:** 3

**Summary:**

The paper presents a novel graph tokenization scheme designed to provide graphs with discrete features as input to standard Transformers. The method involves two main steps:
* First, it employs frequency-guided Eulerian Circuit and Chinese Postman Problem (CPP) approaches to achieve reversible and deterministic serialization of the input graph.
* Second, it uses Byte Pair Encoding (BPE)—a standard in LLMs—to greedily combine frequently occurring token substrings.

The authors validate these results by combining their tokenizer with bidirectional BERT and GTE Transformers, comparing their performance against various GNNs and demonstrating the efficacy of BPE in terms of sequence compression.

**Strengths:**

The proposed approach is elegant and cleanly integrates graph inputs with established LLM best practices. The combination of BPE with graph walks is clever, well-motivated, and useful for inputting graphs directly into Transformers, effectively sidestepping the need for specialized GNN architectures.

The experiments appear strong and thorough. They demonstrate compelling compression rates from BPE and show that the method beats a variety of GNN models, although I note that I lack full context for this specific research area.

The authors are direct about the limitations of their work, specifically regarding continuous features and graph-level tasks, and they suggest reasonable future steps.

**Weaknesses:**

Regarding reversibility: Output sequences consist only of standard labels (Eq. 1), not unique node identifiers. In graphs with many identically labeled nodes (e.g., large carbon lattices), how does the decoder $f^{-1}$ explicitly distinguish between returning to a previously visited node versus arriving at a new node with the same label? Is reversibility guaranteed for all labeled graphs without positional markers? For one particular example, what structure can be accurately recovered from the molecular path in Figure 1?

The serialization example in Figure 1 is confusing. The numbering is difficult to follow (especially as it does not appear to start with 1 as the first edge), and it is unclear why the right-most 'C' appears after 'Br' and 'Cl' in the sequence, given the labeling.

**Questions:**

While the paper mentions generative tasks via GPT-style models as an application, it only evaluates on classification/regression using BERT/GTE. Would pairing the tokenization scheme with autoregressive transformers be able to solve similar supervised graph learning tasks as well (given their general-purpose successes in language modeling)?

It would be interesting to better understand the BPE patterns, lengths, and statistics. Does the learned vocabulary consist mostly of small node/edge tokens, or are there many longer sequence tokens?

---

> ### Author Response · Authors · 2025-11-22
> **Response (1/2)**
>
> We are very grateful for your high praise and insightful questions. Your support is a great encouragement to us.
>
> **Response to Weakness 1: Reversibility with Identically Labeled Nodes**
>
> > "Regarding reversibility: Output sequences consist only of standard labels (Eq. 1), not unique node identifiers..."
>
> Our framework guarantees reversibility up to the graph isomorphism. While the BPE stage is strictly lossless, the serialization stage preserves full topology and labels but does not explicitly encode unique node identifiers. Consequently, the reconstructed graph is structurally identical to the original but may differ in arbitrary node ID assignments. We aim to prioritize structural invariance over administrative indexing. However, for scenarios requiring strict ID preservation, it is feasible to record a node ID-to-token mapping via a side-channel during serialization. We will clarify this distinction and discuss it as a direction for future work in the revision.
>
> **Response to Weakness 2: Clarity of Figure 1**
>
> > "The serialization example in Figure 1 is confusing... numbering is difficult to follow..."
>
> To resolve the visual ambiguity, we have revised **Figure 1** in the updated manuscript. The improved visualization now clearly illustrates the frequency-guided decision mechanism. Specifically, at branching points (e.g., the red node C), the algorithm prioritizes the path associated with the highest-frequency substructure (typically the C-C backbone) before visiting other branches. Specifically, consider the central carbon in an acetic acid molecule, which is connected to a methyl group (via C-C), a carbonyl oxygen (via C=O), and a hydroxyl oxygen (via C-O). At this junction, our algorithm prioritizes the path associated with the highest-frequency substructure (typically the C-C backbone or larger carbon-based groups) before visiting other branches.
>
> **Response to Question 1: Applicability to Autoregressive Models**
>
> > "While the paper mentions generative tasks via GPT-style models as an application, it only evaluates on classification/regression...Would pairing the tokenization scheme with autoregressive transformers be able..."
>
> Our graph tokenizer is model-agnostic and inherently supports autoregressive models for generative tasks. To validate this, we conducted a proof-of-concept experiment in which images were treated as grid graphs and tokenized via our framework. We then trained a standard autoregressive Transformer on MNIST, which successfully generated coherent handwritten digits token-by-token. This result confirms that our approach effectively enables graph generation to be cast as standard sequence modeling. We have included these generation samples in Appendix E.1.

---

> ### Author Response · Authors · 2025-11-22
> **Response (2/2)**
>
> **Response to Question 2: BPE Vocabulary Statistics and Patterns**
>
> > "It would be interesting to better understand the BPE patterns, lengths, and statistics.... Does the learned vocabulary consist mostly of small node/edge tokens...?"
>
> We conducted a detailed statistical analysis of the learned vocabulary ($K=2000$) on the ZINC dataset to address your inquiry. The results, detailed in **Table 1**, demonstrates that **the vocabulary is not dominated by small atomic tokens**. Instead, it consists primarily of composite substructures that effectively encode the graph's semantic topology.
>
> **1. Token Length Distribution: Dominance of Meso-Scale Structures.**
> Our analysis reveals that the vocabulary exhibits a distinct preference for medium-scale patterns. As shown in Table 1, atomic tokens (0-1 nodes) constitute only 7.1% of the vocabulary. In contrast, the distribution peaks in the **4-6 node** range (41.5%), which corresponds to the typical size of molecular functional groups and rings. Combined with the 7-9 node range, over **60%** of the vocabulary represents complex substructures. This distribution proves that BPE successfully identifies an optimal compression level—producing tokens that are large enough to capture structural context (e.g., cycles, branches) yet frequent enough to ensure generalization.
>
> **2. Semantic Pattern Analysis: Autonomous Discovery of Chemical Knowledge.**
> Beyond length statistics, the merging order (Rank) confirms that BPE prioritizes chemically significant substructures purely based on frequency, without explicit supervision. This confirms that our tokenizer effectively "rediscovers" the semantic building blocks of chemistry, encoding them as single discrete tokens to facilitate efficient Transformer processing.
>
> - **Early Discovery of Core Motifs:** Crucial functional groups emerge in the top 5% of the vocabulary (Ranks 0-100). Notably, the Methoxy group (`-OCH3`) is identified at Rank 26, and the complete Benzene ring (`c1ccccc1`, 6 nodes) is included at Rank 40.
> - **Diversity of Chemical Types:** The learned tokens span diverse chemical classes. We observed meaningful Nitrogen-based groups like Amide (`NC=O`, Rank 50) and Nitro (`NO2`, Rank 119), as well as Sulfur-based groups like Sulfonyl (`O=S=O`, Rank 81).
>
> **Table 1:** Fine-grained distribution of token sizes (node counts) in the learned BPE vocabulary on ZINC. The distribution peaks at the 4-6 node range, indicating a preference for functional-group-sized substructures.
>
> | Token Size (Nodes)    | Atomic ($0 \sim 1$) | Small ($2 \sim 3$) | **Medium ($4 \sim 6$)** | Large ($7 \sim 9$) | Huge ($10+$) |
> | :-------------------- | :-----------------: | :----------------: | :---------------------: | :----------------: | :----------: |
> | Vocabulary Proportion |        7.1%         |       28.5%        |        **41.5%**        |       20.4%        |     2.5%     |

---

### Official Review · Reviewer_eXKx · 2025-10-31

**Soundness:** 3
**Presentation:** 2
**Contribution:** 2
**Rating:** 4
**Confidence:** 2

**Summary:**

This paper proposed GraphTokenizer, which essentially just tokenizes the graphs s.t. they can be easily consumed by Transformer based models. The method contains two steps, where the authors first use a deterministic and reversible (in opposite to random walks) sterilization method to transfer the graph data into corpus of sequences, and then use BPE to tokenize them. Once the graphs are tokenized, the following becomes natural as in language modeling. The proposed method performed well across many evaluated benchmarks.

**Strengths:**

1. While the high level idea is not new, the proposed work executed the graph tokenization fairly well, and showcased good performance in the tasks that it can handle.
2. In the evaluations, the proposed method showed good performances as well as efficiency.

**Weaknesses:**

The limitations on page 14 are very on point.

1. As limitations 1&2 mentioned, the focus on only graph-level tasks with discrete features significantly constrains the scope of this work. With such constrains, the proposed method almost only makes sense on protein and chemical graphs, where nodes are atoms, etc.
2. All three limitations kinda showed a theme that this proposed work might not be very suitable for larger graphs such as social networks.

**Questions:**

n/a

---

> ### Author Response · Authors · 2025-11-22
> **Response**
>
> We sincerely appreciate your constructive feedback. We address your specific concerns below.
>
> **Response to Weakness 1: Scope of Applicability**
>
> > "As limitations 1&2 mentioned, the focus on only graph-level tasks with discrete features significantly constrains the scope of this work..."
>
> Regarding limitations on continuous features and node-level tasks, please refer to points 1 and 2 in the **General Response**, where we detail straightforward, lossless extensions to our framework. Regarding graph types, we emphasize that our evaluation already extends well beyond molecular graphs. As shown in Table 4, our experiments cover diverse data domains including academic networks (DBLP), social media (Twitter), synthetic graph theory (COLORS-3, Synthetic), and graphs derived from images (COIL-DEL). These datasets are all integrated into standard libraries like PyG and DGL, demonstrating the method's general applicability across different graph structures.
>
> **Response to Weakness 2: Scalability on Large Graphs**
>
> > "All three limitations kinda showed a theme that this proposed work might not be very suitable for larger graphs..."
>
> We believe that our framework offers a unique advantage for scalability. Unlike classical methods constrained by graph-specific complexity, our approach decouples scalability from the graph model architecture, addressing the challenge through two key mechanisms:
>
> **1. Automatic Scalability via Ecosystem Alignment:**
> Our scalability is driven primarily by aligning graph learning with the broad NLP ecosystem. By transforming the graph scalability challenge into a sequence length problem, our processing capacity expands automatically with advancements in sequence modeling. For example, as context windows have evolved from BERT's 512 tokens to 32k or 1M in modern models, the graph size we can process increases by orders of magnitude without requiring any additional research effort from the graph community.
>
> **2. Practical Capability:**
> By introducing BPE, we substantially compress the graph’s serialized sequence, directly improving scalability. Specifically, with a conservative 10x compression rate from BPE and a standard 10k token context, we can encode graphs with nearly **100k nodes** into a single sequence.
>
> For massive graphs (millions of nodes) where full-graph processing is infeasible for any method, our framework is fully compatible with standard subgraph sampling strategies (e.g., SGFormer). Here, BPE compression allows each sampled subgraph to cover a wider neighborhood than typical GNN hops, and this situation actually falls back to the exact scenario where our method has demonstrated superior performance.
>
> **Clarification on Novelty**
>
> > "While the high level idea is not new..."
>
> We respectfully disagree with the perception that the high-level idea is not new. While the term "graph tokenization" appears in prior literature, it is typically overloaded to refer to **neural encoders for continuous feature alignment** (e.g., GNN projectors) or lossy quantization (e.g., VQ-VAE). In contrast, our work introduces a **discrete symbolic framework**, being the first to adapt Byte Pair Encoding (BPE) to graph data as a deterministic and reversible algorithm. This marks a fundamental shift from model-centric graph adaptation to a **data-centric transformation**, eliminating complex alignment modules and enabling standard Transformers to process graphs directly.

---

### Author Response · Authors · 2025-11-22
**General Response**

We sincerely thank all reviewers for their detailed reviews and invaluable feedback. These constructive comments have been crucial in helping us identify areas for improvement and enhance the quality of our work.

Before addressing each specific concern, we wish to clarify the core contribution of our paper. Our work introduces a paradigm shift in how graph data are interfaced with sequence models. While previous efforts have focused on adapting model architectures to accommodate graph structures (e.g., Graphormer [1]) or aligning graph embeddings with LLMs [2], our approach fundamentally differs. We propose the first framework to adapt Byte Pair Encoding (BPE), a cornerstone of modern LLMs, for graph-structured data.

By converting graphs into discrete token sequences, we completely **decouple graph representation from the model architecture**. It allows the graph learning community to directly leverage the rapidly advancing ecosystem of LLMs without any graph-specific modifications. Any progress in NLP, such as longer context windows or more efficient attention mechanisms, becomes immediately applicable to graphs. We believe that building this seamless bridge is the core value of our work, offering a new and highly scalable path for the field.

Based on your valuable feedback, we first address the common questions raised by several reviewers.

**1. Applicability to Graphs with Continuous Features:**

While our primary focus on discrete labels aligns with BPE's inherent symbolic nature, our framework accommodates continuous features through two strategies. The first involves employing quantization modules (e.g., VQ-VAE) to discretize features into tokenizable codes. Alternatively, a lossless strategy is to inject continuous attributes as **side-channel information** directly into token embeddings, analogous to positional encodings in Transformers. This latter approach accommodates continuous data while effectively preserving the tokenizer’s deterministic and training-free nature. We are actively exploring this direction as future work.

**2. Applicability to Node-level and Edge-level Tasks:**

Our framework is not inherently restricted to graph-level tasks. The current focus on graph-level prediction stems from our use of a BERT-style encoder, which naturally produces a global sequence representation with mean pooling. The key difficulty for node- and edge-level tasks is that BPE may merge a target node into a larger token. A straightforward solution, drawing from established NLP techniques, is to derive a node's representation by pooling the hidden states from the Transformer's output for all tokens containing that node. This method is simple to implement and utilizes the model's contextualized understanding. We intend to explore this direction in future work.

**3. Additional Experiments:**

We fully acknowledge and have acted upon the reviewers' valuable suggestions to include more baselines. A summary of the new comparative experiments and analysis is as follows.

- **Comparison with SOTA Architectures:** We have added comparisons against recent state-of-the-art Graph Transformers and serialization-based methods to benchmark our tokenization approach against the latest advancements.
- **Broader GNN Coverage:** To ensure a comprehensive evaluation across different GNN generations and settings, we have incorporated both classic heterogeneous models and early spectral graph networks.
- **Clarification on Graph Foundation Models (GFMs):** We clarified the distinction between our structural benchmarks and models designed for Text-Attributed Graphs (TAGs). We demonstrate that TAG-oriented models exhibit poor performance when forcibly adapted to non-text structural data due to modality mismatch.
- **Ablation Studies:** We conducted systematic investigations to validate our design choices, including: (1) *Vocabulary Size:* Analyzing the trade-off between compression and performance to validate our choice of $K=2000$; and (2) *Frequency Guidance Unit:* Comparing different granularities (e.g., bigrams vs. trigrams) to confirm that node-edge-node trigrams offer the optimal balance for serialization.

All detailed settings, results, and analysis of these supplementary experiments have been added to Appendix E. We have highlighted all revisions in the updated manuscript in blue for clarity. We will discuss these new results in more detail in our responses to each reviewer.

[1] Ying et al. Do transformers really perform badly for graph representation?

[2] Yu et al. Graph2text or graph2token: A perspective of large language models for graph learning.

---

### Meta-Review · Area_Chair_pZdV · 2025-12-28

**Summary:**

This submission proposes a novel method for converting (discrete) graphs into discrete sequences of tokens, in order to leverage the strong ecosystem for sequential modeling of discrete sequences such as Transformers. The proposed method introduces a new reversible graph serialization algorithm in order to apply BPE, allowing for significant compression of graph motifs. The method as presented is tailored to graphs with discrete labels, and when applicable achieves very strong results over a wide range of baselines on graph-level tasks.

Reviewers raised a number of concerns, which were generally addressed by the author's response.

**Reviewer Concerns:**

Some of the reviewers' concerns include:

**Novelty and positioning**: while some reviewers noted that prior graph tokenization approaches exist, this is the first to apply BPE in a fully discrete setting, as opposed to continuous latents such as VAE-based approaches. The proposed method fully decouples the graph representation problem from any learned neural network, similar to language models.

**Generality**: several reviewers commented on limitations of the method with respect to continuous graph features, which the authors acknowledge in the submission and also provide ideas for circumventing in the author response. A second limitation is to finer-grained manipulation such as node- or edge- level tasks; the authors provide some evidence of the flexibility of the method with additional experiments on MNIST generation with autoregressive Transformers. Although the general concern is not fully addressed (the extensions to continuous features and fine-grained tasks seem plausible but not obvious), we will also point out that a new method does not necessarily have to apply to every case in order to be useful; the proposed approach is the first in a new direction with clear strengths, and more generalizations may be developed over time.

**Baselines**: Several reviewers raised more potential baselines. The authors did a thorough job of including numerous baselines in the response, including GraphGT, Graphormer, FragNet, Graph-ViT-MLPMixer, HAN, and ChebNet; the proposed method still compares favorably.

**Reviewer Scores:**

Reviewer **eXKx** was low confidence; we believe that their concerns were addressed and may have increased their score from 4 to 6.

We believe that **y33R**'s concerns were also adequately addressed and likely could have increased their score from 4 to 6.

---

### Decision · Program_Chairs · 2026-01-26

Accept (Poster)